# TLR induces reorganization of the IgM-BCR complex regulating murine B-1 cell responses to infections

Hannah P Savage[1,2], Kathrin Kläsener[3,4], Fauna L Smith[2,5], Zheng Luo[1], Michael Reth[3,4], Nicole Baumgarth[1,2,5,6]*

[1]Center for Comparative Medicine, University of California, Davis, Davis, United States; [2]Graduate Group in Immunology, University of California, Davis, Davis, United States; [3]BIOSS Centre for Biological Signalling Studies, University of Freiburg, Freiburg, Germany; [4]Department of Molecular Immunology, Institute of Biology III at the Faculty of Biology of the University of Freiburg, Freiburg, Germany; [5]Integrated Pathobiology Graduate Group, University of California, Davis, Davis, United States; [6]Department of Pathology, Microbiology and Immunology, School of Veterinary Medicine, University of California, Davis, Davis, United States

**Abstract** In mice, neonatally-developing, self-reactive B-1 cells generate steady levels of natural antibodies throughout life. B-1 cells can, however, also rapidly respond to infections with increased local antibody production. The mechanisms regulating these two seemingly very distinct functions are poorly understood, but have been linked to expression of CD5, an inhibitor of BCR-signaling. Here we demonstrate that TLR-mediated activation of CD5+ B-1 cells induced the rapid reorganization of the IgM-BCR complex, leading to the eventual loss of CD5 expression, and a concomitant increase in BCR-downstream signaling, both in vitro and in vivo after infections of mice with influenza virus and *Salmonella typhimurium*. Both, initial CD5 expression and TLR-mediated stimulation, were required for the differentiation of B-1 cells to IgM-producing plasmablasts after infections. Thus, TLR-mediated signals support participation of B-1 cells in immune defense via BCR-complex reorganization.
DOI: https://doi.org/10.7554/eLife.46997.001

*For correspondence:
nbaumgarth@ucdavis.edu

**Competing interests:** The authors declare that no competing interests exist.

## Introduction

During lymphocyte development (self)-antigen binding by the TCR and BCR results in negative selection, leading to the removal of strongly self-reactive lymphocytes from the T and B cell repertoire. Depending on the strengths of these antigen-BCR interactions, self-reactive B cells undergo deletion, receptor-editing, or they become anergic, that is unresponsive to antigen-receptor engagement (*Melchers, 2015*).

Self-reactive, anergic bone marrow-derived B cells up-regulate expression of the signaling inhibitor CD5 (*Hippen et al., 2000*). On developing T cells, the levels of CD5 expression correlate with TCR signaling intensity encountered during thymic development, with those most strongly binding to self-antigens expressing the highest levels of CD5 (*Punt et al., 1994*; *Azzam et al., 1998*). While several ligands have been proposed for CD5 (*Punt et al., 1994*; *Berland and Wortis, 2002*; *Biancone et al., 1996*; *Bikah et al., 1998*; *Brown and Lacey, 2010*; *Calvo et al., 1999*; *Pospisil et al., 1996*; *Van de Velde et al., 1991*), none seem to have significant CD5-dependent biological effects. Instead, CD5 expression seems to directly reduce antigen-receptor signaling (*Hippen et al., 2000*; *Punt et al., 1994*; *Azzam et al., 1998*; *Perez-Villar et al., 1999*). Thus, CD5

seems to act primarily as a component of the antigen-receptor complex, directly modulating TCR and BCR signaling.

In addition to anergic B cells, most B-1 cells also express CD5. In contrast to lymphocytes developing postnatal, these primarily fetal and neonatal-derived B cells (*Yuan et al., 2012*; *Zhou et al., 2015*; *Hardy and Hayakawa, 2001*) seem to undergo a positive selection step during development, requiring self-antigen recognition and strong BCR signaling. The lack of self-antigen expression, or the deletion of co-stimulatory molecules that enhance BCR signaling, diminished B-1 cell development, while deletion of negative co-stimulatory signals, or enhanced surface expression of the BCR, resulted in enhanced B-1a cell development (*Berland and Wortis, 2002*; *Hardy and Hayakawa, 2001*; *Casola et al., 2004*; *Hayakawa and Hardy, 2000*; *Nguyen et al., 2017*). Specificity of CD5 + B-1 cells for self-antigens and self-antigen binding during development is consistent with their known self-reactive BCR repertoire (*Hayakawa et al., 1999*; *Lalor and Morahan, 1990*; *Mercolino et al., 1988*; *Yang et al., 2015*) and thus a role for CD5 in silencing B-1 cell responses to BCR-engagement in order to avert autoimmune responses.

Yet, not all B-1 cells express CD5, instead depending on their expression or not of CD5, they are typically divided into two subsets, B-1a and B-1b, respectively. In contrast to B-1b cells, and consistent with their expression of CD5, B-1a cells do not proliferate in response to BCR stimulation (*Morris and Rothstein, 1993*). However, in CD5-/- mice and in mice in which the association of membrane IgM with CD5 was inhibited, mature B-1 cells demonstrated a proliferative response similar to that of conventional B (B-2) cells (*Tarakhovsky et al., 1994*; *Bikah et al., 1996*), further confirming that CD5 expression reduces B-1 cell responsiveness to BCR-signaling.

A BCR-signaling independent response of B-1 cells might be inferred from the fact that B-1 cells strongly respond to innate, TLR-mediated signals, such as LPS, and that they are a major source for 'natural' self-reactive IgM. Moreover, steady-state secretion of natural IgM does not appear to require external antigenic stimulation, as total serum levels of natural IgM and frequencies of IgM-secreting B-1 cells are similar in mice held under both, SPF and germfree housing conditions (*Baumgarth et al., 2015*; *Ochsenbein et al., 1999*). However, natural IgM production is not stochastic, but instead likely driven by expression of self-antigens. This was demonstrated by Hayakawa et al, who showed a lack of anti-Thy-1 self-reactive IgM antibodies in the serum of Thy-1-deficient but not Thy-1 expressing mice (*Hayakawa et al., 1999*; *Hayakawa et al., 2003*), as well as repertoire studies by Yang et al, which showed selective and extensive clonal expansion of certain CD5 + B-1 cell clones during the first 6 months of life, including in germfree mice (*Yang et al., 2015*).

Furthermore, B-1 cells can be also actively involved in immune responses to various pathogens (*Haas et al., 2005*; *Alugupalli et al., 2003*; *Alugupalli et al., 2004*; *Gil-Cruz et al., 2009*; *Choi and Baumgarth, 2008*). Given that CD5 is a BCR inhibitor, it was suggested that CD5- B-1b cells, but not B-1a cells, respond to pathogen encounters in an antigen-dependent manner. Haas and colleagues, conducting studies in CD19-deficient mice that lack B-1a development, concluded that B-1a cells are responsible for natural IgM secretion, while only the B-1b cells responded to *Streptococcus pneumonia* infection (*Haas et al., 2005*). Similarly, CD5- B-1b cells were shown to expand and secrete protective IgM after infection with *Borrelia hermsii* and *Salmonella typhimurium* (*Alugupalli et al., 2003*; *Alugupalli et al., 2004*; *Gil-Cruz et al., 2009*).

This model of a 'division of labor' between B-1a and B-1b cells leaves the B-1 cell response to influenza infection as an outlier. Chimeric mice reconstituted with either allotypically-marked CD5 + or CD5- B-1 cells showed that only CD5+ B-1 cells were responding in vivo to influenza infection with migration from the pleural cavity to the draining mediastinal lymph nodes (MedLN) in a Type I IFN-dependent process, where they differentiated into IgM-secreting cells (*Choi and Baumgarth, 2008*; *Waffarn et al., 2015*). The reasons for the apparent different behaviors of CD5+ and CD5- B-1 cells in the various infectious disease models are unexplained. Furthermore, it is unclear how B-1 cells expressing CD5 can participate in antigen-specific immune responses.

This study addresses some of these questions and reconciles previous divergent findings on B-1 cell responses to infections by demonstrating that only CD5+ B-1 cells respond to influenza virus as well as *Salmonella* infections, but that once activated, these B-1 cells lose expression of CD5 and thus become 'B-1b' like. Mechanistically, the downregulation of CD5 requires expression of TLR, triggering of which resulted in the reorganization of the IgM-BCR complex. BCR reorganization led to the rapid dissociation, and then eventual loss of CD5 from the complex, and triggered enhanced IgM-CD19 and CD79:Syk interactions, resulting in enhanced down-stream BCR-signaling. Thus, TLR-

mediated signals support participation of B-1 cells in immune defense via BCR-complex reorganization, linking innate and adaptive antigen-recognition by B-1 cells.

## Results

### CD5 negative B-1 cells are responsible for local IgM secretion after influenza infection

We previously identified three populations of cells involved in natural IgM secretion: CD5+ B-1 cells, CD5- B-1 cells, and plasma cells, the latter are CD19- and CD138/Blimp-1+ (*Savage et al., 2017*) and also B-1-derived (B-1PC) (*Savage et al., 2017*). This was shown using a 'neonatal chimera' model, in which host B-1 cells are replaced in neonatal host mice by congenic but Ig-allotype-disparate donor B-1 cells, while the host B-2 cells remain of the host and thus its allotype (*Lalor et al., 1989*). After full reconstitution B-1 cells as well as their secreted IgM can be identified and quantified using allotype-specific anti-IgM (and anti-IgD) antibodies. Because B-1-derived IgM is important for protection from lethal influenza infection (*Baumgarth et al., 2000*), we sought to determine which B-1 cell populations generate IgM in the draining (mediastinal) lymph nodes (MedLN) after influenza infection (*Choi and Baumgarth, 2008*).

Examination of the MedLN of neonatal chimeras showed that B-1 cells migrated to MedLN and then rapidly differentiated to IgM-secreting B-1PC on day seven after infection with influenza A Puerto Rico 8/34 (A/PR8) (*Figure 1A*). Neonatal chimeric mice generated with B-1 donor cells from Blimp-1 YFP reporter mice (*Fooksman et al., 2010*; *Rutishauser et al., 2009*) confirmed the presence of Blimp-1-YFP+ B-1PC in the MedLN (*Figure 1B*). The MedLN B-1PC mostly lacked expression of CD5, particularly among the Blimp-1$^{hi}$ cells (*Figure 1C*). Also, the CD5+ Blimp-1-YFP+ cells expressed less Blimp-1-YFP than the CD5- Blimp-1-YFP+ B-1 cells (*Figure 1C*, left). The data were unexpected, as we had shown previously that only the CD19+CD43+CD5+but not the CD5- B-1 cells were able to migrate from the pleural cavity to the MedLN after influenza infection, where they differentiated into IgM-secreting cells (*Choi and Baumgarth, 2008*; *Waffarn et al., 2015*).

To investigate the contribution of CD5- B cells to local IgM secretion, we FACS-sorted CD19 +IgM+IgD$^{lo}$CD43+ CD5+ and CD5- B cells on days 3, 5, and seven after influenza infection from C57BL/6 mice (*Figure 1D*), which were then cultured for 2 days to analyze spontaneous IgM secretion by ELISA. Consistent with the presence of CD5- B-1PC, the CD5- cells secreted more IgM compared to the CD5+ cells, measuring total IgM concentrations and calculating IgM production per cell (*Figure 1E*). Sorted CD5+ cells did not secrete measurable amounts of IgM unless harvested after day 5 of infection.

Because CD5- B-1 cells and IgM-secreting B-2 derived plasmablasts express a similar phenotype (IgM+ IgD- CD5- CD19+ CD43+), the CD5- cells could have contained both B-1 cells and/or B-2-derived IgM-secreting cells. To determine the contribution of CD5- B-1 cells to IgM secretion in the MedLN, we infected allotype-disparate B-1 cell neonatal chimeras, in which B-1 (Igh$^b$) and B-2 (Igh$^a$) cells and their secreted antibodies can be distinguished based on Igh-allotype differences (*Lalor et al., 1989*). The studies confirmed our previous findings that among CD19$^{hi}$IgM$^b$+-IgD$^{lo}$CD43+B-1 cells in the MedLN, about 70% expressed CD5 after influenza infection (*Figure 1F– G*).

Because we had shown previously that Blimp-1+ B-1PC have reduced or absent CD19-expression (*Savage et al., 2017*) and found here that these cells are present after influenza infection (*Figure 1A–B*) and often lacked CD5-expression (*Figure 1C*), we expanded the analysis to include all IgM$^b$-expressing (B-1 donor-derived) and IgM$^a$ negative (recipient-derived) cells, regardless of expression of CD19 or other surface markers (*Figure 1H*). Of note, a small number of double positive (IgM$^a$+IgM$^b$+) cells were always observed but excluded from the analysis, as these are likely IgM$^a$+ B-2 cells with surface-bound serum IgM (B-1-derived and thus IgM$^b$), attached via surface FcµR (*Nguyen et al., 2017*) (*Figure 1A/H*). In contrast to the analysis described above, this expanded analysis of all B-1 donor Igh-b cells revealed that the frequency of CD5 negative MedLN B-1 cells increased after influenza infection (*Figure 1H–I*), consistent with the development of CD5- B-1PC in this compartment (*Figure 1A–C*). Furthermore, FACS-sorting and culture of CD5+ and CD5- B-1 cells showed that a higher frequency and total number of CD5- B-1 cells secreted IgM in the MedLN compared to CD5+ B-1 cells on days 3, 5, and seven after infection (*Figure 1J*). Thus,

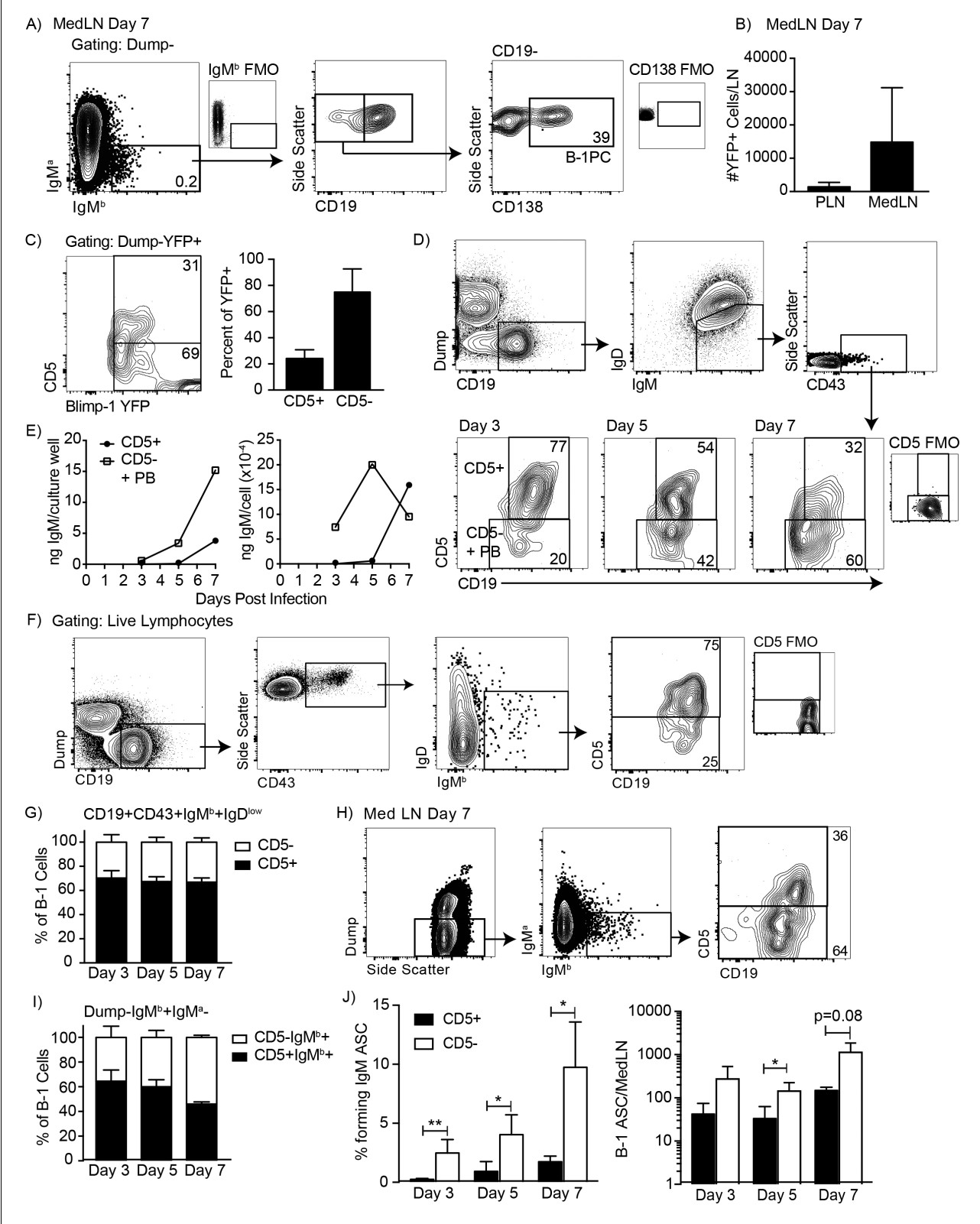

**Figure 1.** CD5 negative B-1 cells secrete most IgM in the mediastinal lymph nodes (MedLN) after influenza infection. (**A**) FACS plot of MedLN cells from day seven influenza-A/PR8-infected neonatal chimeric mice generated with Ighb B-1 donor cells and Igha host cells. Shown is gating to identify IgMb+CD19+B-1 cells and IgMb+CD19 CD138+B-1PC. FMO, 'fluorescence minus one' control stains. (**B**) Mean number ± SD of Blimp YFP+ cells in peripheral LN (PLN) and MedLN of day seven influenza-infected neonatal chimera generated with B-1 donor cells from Blimp-1-YFP mice (n = 4). (**C**)
*Figure 1 continued on next page*

*Figure 1 continued*

FACS plot (left) and (right) mean percentage ± SD of CD5+ and CD5- cells among total Blimp-1 YFP+ cells (n = 13). (D) FACS gating strategy for sorting CD19+IgM+IgD$^{lo}$CD43+CD5+ or CD5- cells in the MedLN on days 3, 5, or seven after influenza infection of C57BL/6 mice, pooled from n = 2–3 per time point. (E) Concentration (ng/ml) IgM in supernatant (left) and secreted (ng x 10$^{-4}$) per cell (right) of sorted cells measured by ELISA. (F) FACS gating strategy and (G) mean percentage ± SD of CD19+CD43+IgM$^{b}$+IgD$^{lo}$ and CD5+ or CD5- B-1 cells among at indicated times after infection (n = 6–7 per time point). (H–I) Samples from (G) regated to show total B-1 populations. (H) Sample FACS plot and (I) percentage ± SD of CD5+ and CD5- B-1 cells among total (Dump- IgMb+ IgMa-) B-1 cells at indicated times after infection (n = 6–7 per time point). (J) Mean percentage ± SD (left) and total number ± SD (right) of FACS-sorted CD5+ and CD5- B-1 cells (IgM$^{b}$+IgM$^{a}$-) that formed IgM antibody-secreting cells (ASC) in each MedLN, as measured by ELISPOT (n = 3–4 per time point). Results are representative of >4 (A), 3 (B), and 2 (F, I), or are combined from 2 (D, E, G, H) or 3 (C) independent experiments. Values in (I) were compared by unpaired Student's t test (*=p < 0.05, **=p < 0.005).
DOI: https://doi.org/10.7554/eLife.46997.002

CD5- B-1 cells increase in the MedLN and are a major source of local IgM production after influenza infection.

## CD5+ B-1 cells decrease CD5 expression after LPS stimulation in vitro

To reconcile our previous findings about the role of CD5+ B-1 cells in influenza infection (*Choi and Baumgarth, 2008*; *Waffarn et al., 2015*), we considered whether CD5 surface expression may change after B-1 cell activation. Indeed, approximately 40% of highly purified FACS-sorted CD5+ B-1 cells from the peritoneal cavity lacked CD5 expression when cultured for 3 days in the presence but not absence of LPS, a stimuli that is known to induce IgM production by body cavity B-1 cells (*Su et al., 1991*) (*Figure 2A*). CD5 surface expression was unaffected during the first 2 cell divisions following stimulation, but was then quickly lost during the next 1–2 divisions (*Figure 2B*). Both, surface-expressed CD5 and *cd5* mRNA, as assessed by qRT-PCR, were decreased among B-1 cells after 3 days of LPS stimulation (*Figure 2C–D*). Surface CD5 levels were decreased by 1.5 days of culture, while *cd5* mRNA was not reduced until 2 days after culture onset (*Figure 2C–D*). The stimulated cells began secreting IgM before CD5 levels were reduced, but the increase in IgM secretion was more pronounced after 2 days of stimulation compared to the earlier time points (*Figure 2E*).

A number of control studies were performed to ensure that the reduced frequencies of CD5+ B-1 cells in the cultures were not due to selective expansion of small numbers (<5%) of CD5- cells that might have contaminated the cultures at onset. First we separated CD5+ and CD5- B-1 cells from the body cavities by FACS to very high purities, and then cultured pure (100%) CD5+ B-1 cells, as well as cultures of CD5+ B-1 cells to which we added 1% and 5% CD5- B-1 cells, respectively. The data showed that the frequencies of CD5+ and CD5- cells after 3 days of culture were unaffected by the initial composition of the culture wells (*Figure 2—figure supplement 1A*). There was no significant difference in either CD5 MFI or in the percent of CD5+ and CD5- cells on day 3 of culture, independent of whether small numbers of CD5- cells were added to the CD5+ B-1 cell cultures (*Figure 2—figure supplement 1B*). Thus, small percentages of CD5- B-1 cells at culture onset, representative of potential sort impurities, could not explain the lack of CD5 expression by the CD5+ B-1 cells stimulated with LPS for 3 days.

Next, we compared the ability of CD5+ and CD5- B-1 cells to survive and/or proliferate with and without LPS stimulation to ensure that CD5- B-1 cells do not demonstrate a more robust response than CD5+ B-1 cells. To ensure that the two populations were exposed to the same culture conditions, CD5+ and CD5- B-1 cells were sorted, labeled with different proliferation dyes, and cultured together (*Figure 2—figure supplement 1C*). Compared to B-1 cells that expressed CD5 on day 0, CD5- cells did neither demonstrate better survival (*Figure 2—figure supplement 1D*) nor enhanced proliferation in response to LPS stimulation compared to the CD5+ cells, in terms of both, the percentage of cells that underwent division, as well as the numbers of divisions each B-1 cell underwent (*Figure 2—figure supplement 1E–F*). In fact, the CD5+ B-1 cells had better overall survival rates compared to CD5- B-1 cells. Reflecting the similar rates of proliferation and the increased survival of the CD5+ B-1 cells, populations of B-1 cells that expressed CD5 at culture onset were present at higher frequencies of input cells compared to B-1 cells that were CD5 negative (*Figure 2—figure supplement 1G*). We conclude that CD5+ B-1 cells lose CD5 surface and mRNA expression after in vitro LPS stimulation.

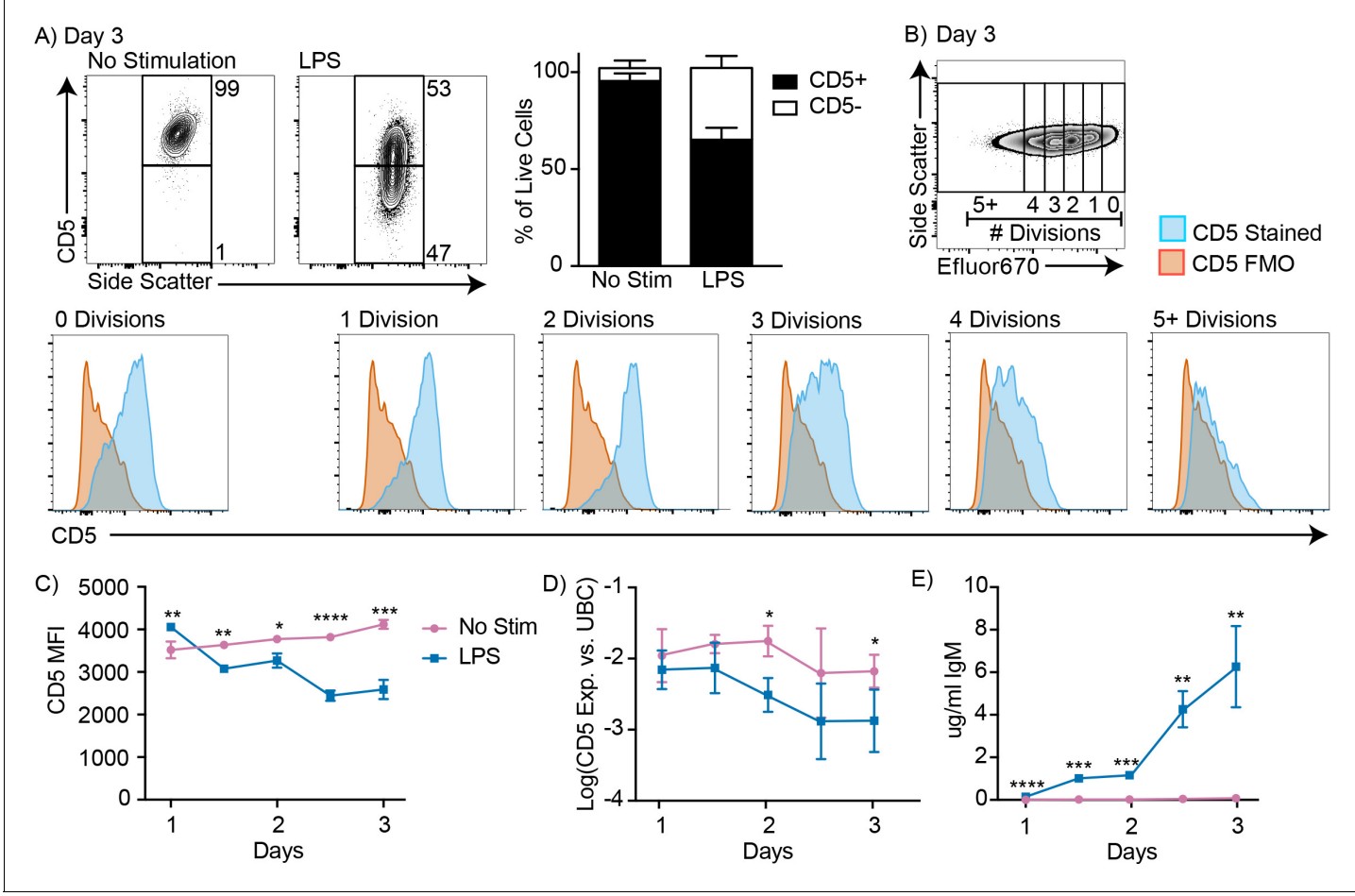

**Figure 2.** CD5+ B-1 cells decrease CD5 expression after LPS stimulation in vitro. (A) Representative FACS plots (left) and mean percentage ± SD (right) of CD5+ and CD5- B-1 cells after FACS-purified peritoneal cavity CD19+ CD23- CD5+ B-1 cells were cultured with or without 10 μg/ml LPS for 3 days (n = 18). (B) CD5 expression on FACS-purified Efluor 670-stained proliferating peritoneal cavity CD5+ B-1 cells stimulated with LPS compared to CD5 FMO (fluorescence minus one) control. (C) Mean CD5 MFI ± SD, determined by flow cytometry, (D) mean Log(*cd5* mRNA expression) ± SD, determined by qRT-PCR, and (E) mean IgM secretion ± SD (μg/ml), determined by ELISA, after purified peritoneal cavity CD5+ B-1 cells were cultured for indicated times with LPS (n = 3–4 per time and data point). Results are combined from 4 (A), or are representative of >5 (B), and 2 (C-E) independent experiments, respectively. Values in (C–E) were compared using an unpaired Student's t test (*=p < 0.05, **=p < 0.005, ***=p < 0.0005, ****=p < 0.00005).

DOI: https://doi.org/10.7554/eLife.46997.003

The following figure supplement is available for figure 2:

**Figure supplement 1.** CD5- B-1 cells do not survive better or proliferate more compared to CD5+ B-1 cells.
DOI: https://doi.org/10.7554/eLife.46997.004

## CD5+ B-1 cells differentiate into CD5- IgM secreting cells after stimulation with multiple TLR agonists

Endosomal TLR agonists Imiquimod (TLR7) and ODN CpG 7909 (TLR9) also induced CD5 downregulation on CD5+ B-1 cells after 3 days of culture (*Figure 3A*), as did stimulation with lipids from *Mycobacterium tuberculosis* (Mtb lipids), which activate cells primarily through TLR2 (*Basu et al., 2012*) (*Figure 3C*). These findings are consistent with the loss of CD5 expression seen after stimulation with various TLR agonists by *Kreuk et al. (2019)*. Similar to LPS stimulation, CD5 expression decreased as the cells divided (*Figure 3B*). In contrast, stimulation of CD5+ B-1 cells isolated from mice lacking all TLR-signaling due to a deletion of genes encoding Unc93, TLR2 and TLR4 (kind gift of Greg Barton, UC Berkeley), were unable to respond with proliferation (*Figure 3D*), and failed to

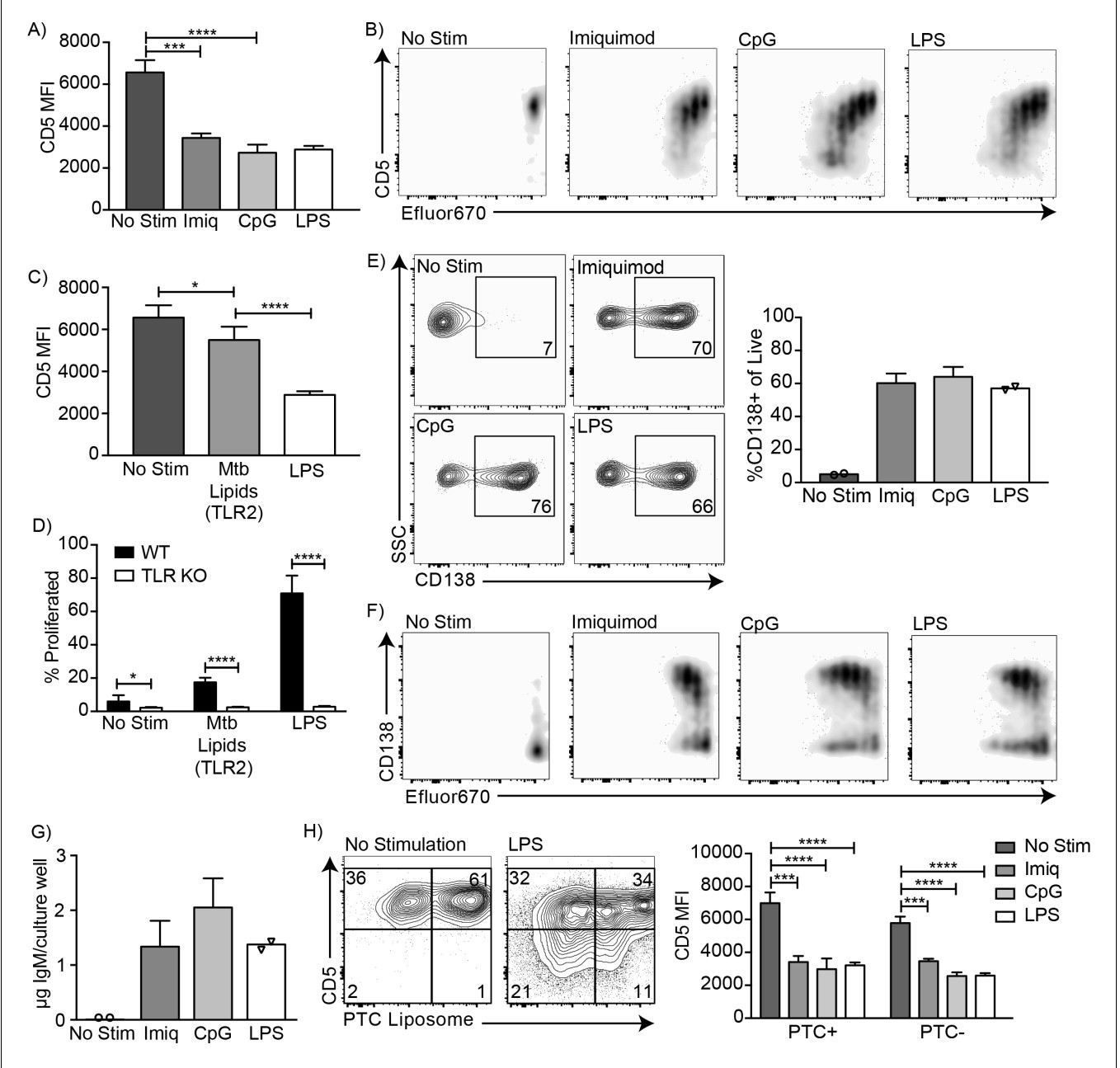

**Figure 3.** CD5+ B-1 cells differentiate into CD5- IgM secreting cells after TLR-mediated activation. (**A**) CD5 MFI ± SD and (**B**) representative FACS plots for CD5+ B-1 cells cultured without stimulation or with Imiquimod (TLR7 agonist), CpG 7909 (TLR9 agonist), or LPS (TLR4 agonist) (n = 3–5). (**C**) Mean CD5 MFI ± SD of CD5+ B-1 cells cultured without stimulation or with *Mycobacterium tuberculosis* (Mtb) lipids (TLR2 agonist) or LPS (n = 4–5). (**D**) Mean percentage ± SD of B-1 cells from wild type (WT) or Tlr2$^{-/-}$xTlr4$^{-/-}$xUnc93b1$^{3d/3d}$ (TLR KO) mice that underwent at least one division after culture without stimulation or stimulated with *Mycobacterium tuberculosis* (Mtb) lipids or LPS (n = 6–9 per group). (**E**) FACS plots (left) and mean percentage ± SD (right) of CD138+ cells, and (**F**) representative FACS plots for CD138 expression among proliferating cells. (**G**) Mean IgM concentration ± SD (µg total per culture well) of cultured CD5+ B-1 cells stimulated or not with Imiquimod (TLR7 agonist), CpG 7909 (TLR9 agonist), or LPS (TLR4 agonist) (n = 2 for no stimulation and LPS, n = 5 for Imiquimod and ODN). (**H**) Sample FACS plot (left) and mean CD5 MFI ± SD (right) of PTC liposome-binding (PTC+) and non-PTC liposome-binding (PTC-) cells for CD5+ B-1 cells cultured without stimulation or with Imiquimod (TLR7 agonist), CpG 7909 (TLR9 agonist), or LPS (TLR4 agonist) (n = 3–5). Results are combined from two (**D, E–G**), or are representative of three (**A**) or two (**B, C, H**) independent experiments, respectively. Values compared in (**A, C–D**) using an unpaired Student's t test (*=p < 0.05, **=p < 0.005, ***=p < 0.0005, ****=p < 0.00005).

DOI: https://doi.org/10.7554/eLife.46997.005

lose CD5 (not shown). Thus, B-1 cells were stimulated via TLR-engagement and not via the BCR. In all instances, the loss of CD5 was correlated with the differentiation of CD5+ B-1 cells to IgM-secreting cells, as stimulation of these cells with Imiquimod, CpG, and LPS for 3 days resulted in increased percentages of CD138+ cells (*Figure 3E–F*) and an increase in IgM concentrations in the culture supernatants (*Figure 3G*).

Finally, we examined whether phosphatidylcholine (PTC)-binding B-1 cells can lose CD5 surface expression after TLR-stimulation. PTC is a well-known specificity of a large subset of peritoneal cavity CD5+ B-1 cells (*Mercolino et al., 1988*; *Arnold and Haughton, 1992*). PTC-binding B-1 cells, identified by incubation of cells with a fluorescent PTC-liposome (kind gift of A. Kantor, Stanford University), lost CD5 expression similarly to PTC non-binders (*Figure 3H*). We conclude that TLR-mediated stimulation of CD5+ B-1 cells ('B-1a') causes the loss of CD5 surface expression, making these cells phenotypically indistinguishable from the proposed 'sister' B-1 cell population, the CD5- 'B-1b' cells.

## CD5+ B-1 cells become CD5- IgM ASC in the MedLN after Influenza infection

These results raised the possibility that pleural cavity CD5+ B-1 cells respond to influenza virus infection with down-regulation of CD5 in the MedLN. The data would explain the increases in CD5- B-1 cells in the MedLN after influenza infection (*Figure 1*). They would also explain how the frequencies of CD5- B-1 cells increased at that site, despite the fact that we had shown previously with neonatal chimeras reconstituted with only CD5- B-1 cells that CD5- B-1 cells cannot enter the MedLN after influenza infection, and that the CD5+ B-1 cells were sufficient to induce the entire B-1 cell response (*Choi and Baumgarth, 2008*).

To confirm and expand these data and because these previous studies showed that only CD5+ B-1 cells could enter the MedLN, we established neonatal chimeras with varying mixes of CD5+ and CD5- B-1 cells (*Figure 4A*) and tested whether we could see a correlation between the frequencies of CD5+ and CD5- cells in the MedLN and/or the levels of B-1-derived IgM secretion in MedLN after influenza infection. For clarity we binned the results into chimeras reconstituted with >50% CD5- B-1 cells,>50% CD5+B-1 cells or only CD5+ B-1 cells (*Figure 4B*). As shown previously (*Choi and Baumgarth, 2008*), MedLN of mice reconstituted with mostly CD5- B-1 cells had reduced MedLN B-1 cell after infection (*Figure 4B*). Of note, among the total donor B-1 cells in the MedLN, the frequencies of CD5+ and CD5- cells were similar, regardless of the initial percentage of CD5+ cells (*Figure 4C–D*), consistent with the CD5+ B-1 cells losing surface CD5 expression. This is further consistent with the fact that in chimeras generated with only CD5+ B-1,>50% of B-1 cells in the MedLN lacked CD5 expression (*Figure 4D*).

Allotype-specific ELISPOTs showed that chimeric mice generated with only CD5+ B-1 cells formed significantly higher frequencies of B-1-derived IgM-secreting cells compared to chimeric mice generated with predominantly CD5- B-1 cells (*Figure 4E*). In fact, chimeras generated with CD5- B-1 cells showed no more B-1 derived IgM ASC in their MedLN than uninfected chimeras, consistent with their previously reported deficiency in entering the MedLN after infection (*Figure 4E*, left panel) (*Waffarn et al., 2015*; *Choi and Baumgarth, 2008*). There was a significant positive correlation between the frequencies of CD5+ B-1 cells transferred to generate the neonatal chimeras and the ability of the mice to generate IgM ASC following influenza virus infection (*Figure 4E*, right panel).

CD5+ B-1 cells failed to show signs of clonal expansion following their accumulation in the MedLN (*Choi and Baumgarth, 2008*), which was confirmed using BrdU injection on day six after infection. However, the CD5- MedLN B-1 cells showed increased proliferation compared to their counterparts in body cavities (*Figure 4F*). Among B-1 cells the proliferation rate was highest among the CD138+ B-1PC, with rates similar to that of the B-2 CD138+ plasma cell compartment (*Figure 4F*). The data support the hypothesis that CD5- B-1 cells, and in particular B-1PC in the MedLN, arise from proliferating CD5+ pleural cavity B-1 cells that accumulate in the MedLN and differentiate into CD5- IgM ASC following acute influenza virus infection.

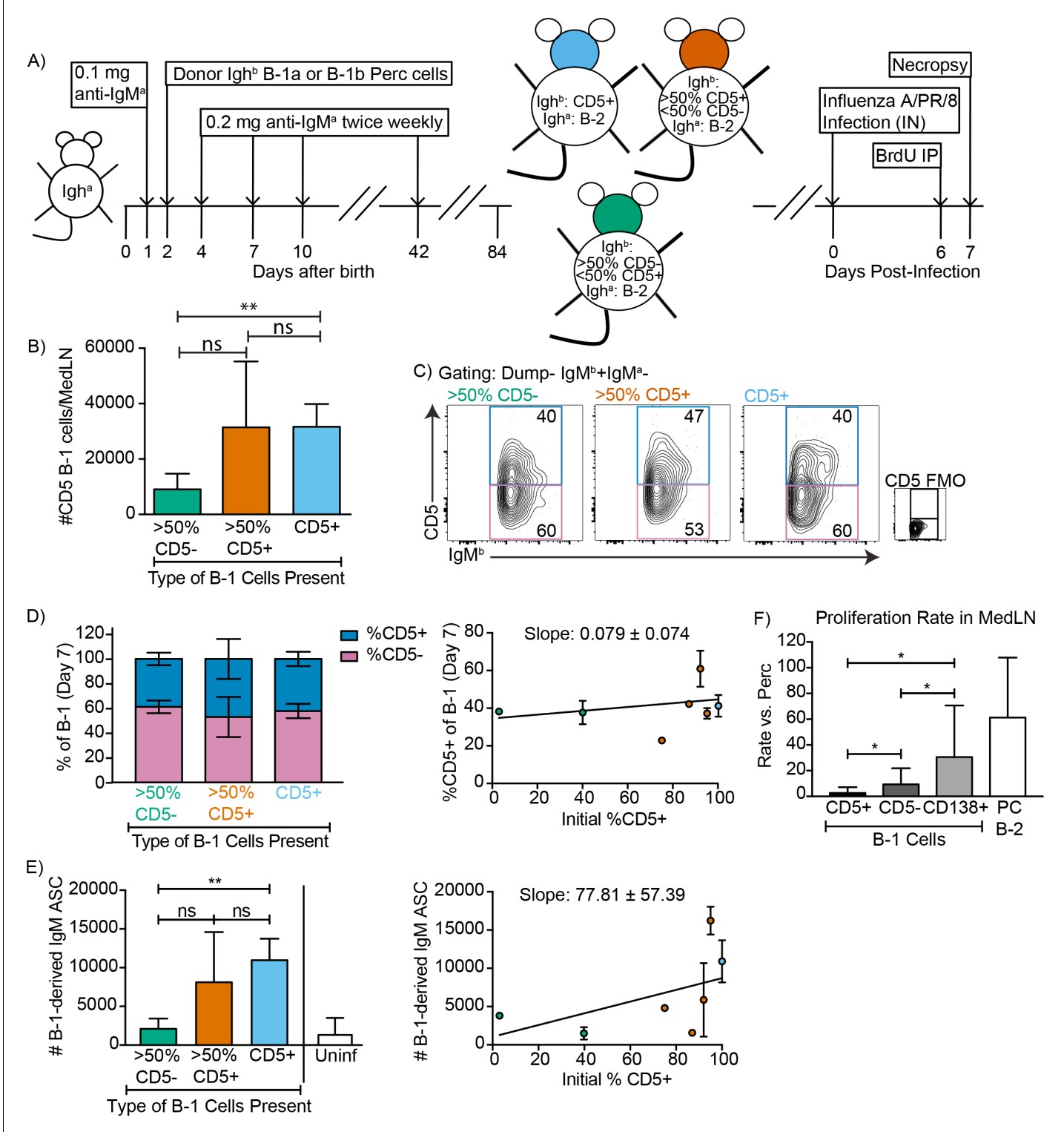

**Figure 4.** CD5+ B-1 cells differentiate to CD5- IgM ASC in the MedLN after Influenza infection. (A) Neonatal chimeric mice were generated with FACS sorted CD19+ CD23- Ighb+ CD5+ (100%, blue), mostly CD5+ (orange), or mostly CD5- (green) peritoneal cavity-derived B-1 cells and infected with influenza A/Puerto Rico 8/34 for 7 days. (B) Mean number ± SD of B-1 cells in the MedLN of mice 7 days after infection. (C) FACS plot and (D, left) mean percentage ± SD of Dump- IgMb+ IgMa CD5+ and CD5- MedLN B-1 cells on day 7. CD5 FMO (fluorescence minus one) control for CD5. (D, right) Mice were grouped by initial percentage of CD5+ and CD5- B-1 cells (left) and % MedLN CD5+ B-1 cells present on days 0 (initial %) and 7 of infection were plotted with a line of best fit. (E) Mean B-1 derived IgM ASC ± SD per MedLN, grouped by initial percentage of CD5+ and CD5- cells

*Figure 4 continued on next page*

Figure 4 continued

(left) and plotted based on initial starting percentage of CD5+ cells (right) with a line of best fit. (F) Mean proliferation rate per day ± SD of CD5+, CD5-, and CD138+ B-1 cells and CD138+ B-2 cells (B-2 PC) in the MedLN of infected chimeras compared to proliferation rate per day of similar populations (B-1 or B-2 cells) in the peritoneal cavity of each mouse as determined by BrDU incorporation. Results for infected mice in (B–F) are combined from four independent experiments (n = 4 for>50% CD5-, n = 7 for>50% CD5+ cells, n = 5 for pure CD5+ cells). Results for uninfected chimeras in (E) are combined from three independent experiments, n = 6. Values in (B, D–F) were compared by unpaired Student's t test (*=p < 0.05, **=p < 0.005).
DOI: https://doi.org/10.7554/eLife.46997.006

The following figure supplement is available for figure 4:

Figure supplement 1. CD5+ B-1 cells differentiate to CD5- IgM ASC in the MesLN and Peyer's patches after *S. typhimurium* infection.
DOI: https://doi.org/10.7554/eLife.46997.008

## CD5+ B-1 cells become CD5- IgM ASC in the Mesenteric LNs and Peyer's Patches after Salmonella typhimurium infection

Numerous infection models have reported CD5- 'B-1b' cell responses after infection, including studies on mice infected with *Streptococcus pneumonia* (*Haas et al., 2005*) and *S. typhimurium* (*Gil-Cruz et al., 2009*). This has led to the concept that the CD5- B-1b are a 'responder' B-1 cell population, whereas CD5+ B-1 cells generate natural IgM exclusively in the steady state (*Haas et al., 2005*; *Alugupalli and Gerstein, 2005*). We therefore aimed to reexamine whether activation and differentiation of CD5+ B-1 cells into CD5- IgM ASC were more universal outcomes of CD5+ B-1 cell activation to infections.

Neonatal chimeric mice generated with varying ratios of FACS-purified CD5+ and/or CD5- B-1 cells, as described above were orally infected with *S. typhimurium* (*Figure 4—figure supplement 1A*). Consistent with the studies of MedLN B-1 cell populations after influenza infection, we found an increased number of B-1 cells in the infected Mesenteric LN (MesLN) of mice given CD5+ B-1 cells vs. those given CD5- B-1 cells (*Figure 4—figure supplement 1B*, left). A similar trend was seen in the Peyer's Patches, but did not reach statistical significance, likely due to the very small number of total B-1 cells in that tissue (*Figure 4—figure supplement 1B*, right). Also consistent with studies on the MedLN after influenza infection, we found a similar percentage of CD5+ and CD5- B-1 cells (identified as IgM$^b$+IgM$^a$-) in the MesLN and Peyer's Patches on day four after infection, regardless of the initial percentage of CD5+ cells used to reconstitute the B-1 compartment of these mice . We again found that > 50% of the B-1 cells in tissues of chimeras established with only CD5+ B-1 cells lacked CD5 surface expression (*Figure 4—figure supplement 1C* and these chimeras were the most competent at forming IgM secreting cells after infection (*Figure 4—figure supplement 1D*). In contrast, the MesLN and PP of chimeric mice that were given primarily B-1b cells formed fewer IgM secreting cells, although more than the uninfected chimeras . Although we did see a significant decrease in the number of B-1 cells in the MesLN of mice given 95% CD5+ B-1 cells compared to those receiving 100% CD5+ *Figure 4—figure supplement 1B*, left), this is unlikely of biological significance, since there is similar IgM secretion in the MesLN between these two groups of mice, *Figure 4—figure supplement 1B*, left).

The *S. typhimurium* surface antigen OmpD had been reported previously to stimulate IgM secretion exclusively by CD5- 'B-1b' cells (*Gil-Cruz et al., 2009*). However, in our hands, although total B-1-derived IgM was increased after infection in the MesLN (*Figure 4—figure supplement 1D*) OmpD-specific B-1-derived IgM ASC in the MesLN did not increase significantly (*Figure 4—figure supplement 1E*). Instead, we found OmpD-specific IgM secretion only by host-derived, thus B-2 plasmablasts. Of note, the phenotype of activated B-2-derived plasmablasts is indistinguishable from that of the so-called 'B-1b' cells (CD19$^{lo}$ CD5- CD45R$^{lo}$ IgM+ CD43+ (*Figure 4—figure supplement 1F*) and thus only identifiable using a lineage-marking approach, such as used here.

Together these findings demonstrate that B-1 cells accumulate in draining lymph nodes at the site of both, bacterial and viral infections, where they lose CD5 expression and become the main source of B-1 derived secreted IgM. In vitro this process is recapitulated by stimulation with various TLR-ligands.

## Changes in BCR signaling following innate activation of B-1 cells

Surface CD5 expression by B-1 cells has been linked previously to their inability to proliferate in response to BCR-mediated signaling (*Bikah et al., 1996*). To analyze the association of CD5 with the BCR on B-1 cells in steady state and to determine what changes the stimulation of the BCR may induce on B-1 cells, we analyzed the IgM-BCR-complexes on the cell surface of highly FACS-purified, then rested, peritoneal cavity CD5+ CD45 R$^{lo}$ CD23- B-1 and splenic CD45R$^{hi}$ CD23+ CD5 follicular B cells using Proximal Ligation Assay (PLA). On B-1 cells, both CD19 and CD5 were strongly associated with the surface-expressed IgM-BCR, while CD5 was not directly associated with the co-stimulator and signaling molecule CD19 (*Figure 5A*). Consistent with the lack of stimulation and strong interaction between IgM and CD5, the BCR-signaling chain CD79 only weakly interacted with the adaptor molecule Syk in B-1 cells in the steady-state. B-2 cells lack CD5 expression, and CD19 did not interact with the IgM-BCR prior to stimulation (*Figure 5A*).

Stimulation of B-1 and B-2 cells with CpG induced strong proliferation of both cell populations (*Figure 5B*). As expected, and although anti-IgM induced strong proliferation by B-2 cells, CD5+ B-1 cells failed to respond to the same stimulus (*Figure 5B*). The lack of responsiveness of the CD5+ B-1 cells to BCR-stimulation was explained by the PLA data, which showed the maintenance and even increase in IgM-BCR:CD5 association and an increase in the association of the inhibitor CD5 with CD19 following anti-IgM treatment. Furthermore, B-1 cells lost the interaction of the IgM-BCR with CD19. Consequently, CD79-Syk interactions remained very low (*Figure 5C*). Thus, BCR-engagement on B-1 cells inhibits BCR-signaling by reducing steady-state IgM-CD19 interactions and likely also by initiating instead interactions between CD5 and CD19.

In contrast to direct stimulation of the IgM-BCR, CpG stimulation led to changes in the BCR-signaling complex that are consistent with induction of positive BCR-signaling, and/or the ability to signal through the antigen-receptor (*Figure 6*). CpG stimulation strongly increased the interaction of IgM-BCR:CD19 and reduced CD5:IgM-BCR proximity. The dissociation between the IgM-BCR and CD5 occurred within 24 hr of stimulation (*Figure 6A*) and thus significantly before the reduction in surface CD5 expression was measurable by flow cytometry (*Figure 2C*) and before changes to CD5 transcription (*Figure 2D*). The already weak CD5:CD19 interaction was further reduced (*Figure 6A*), consistent with the eventual loss in surface CD5 expression noted following stimulation (*Figure 3*). These rapid changes in the BCR-signaling complex were associated with increases in CD79:Syk interaction, suggesting active BCR-signaling in CpG-stimulated B-1 cells (*Figure 6A*). This was further supported by sustained increased levels of phosphorylated Akt (pAkt; pS473) following stimulation of CD5+ B-1 cells with CpG, while anti-IgM stimulation reduced pAkt levels below that of unstimulated B-1 cells by 24 hr, after an initial increase (*Figure 6B*). We also noted increased Nur77 expression in CpG- but not anti-IgM-stimulated B-1 cells, further suggesting BCR-signaling is linked to CpG-stimulation after 24 hr (*Figure 6C*), but not after 3 hr (not shown). Given that B-1 cell responses following CpG-stimulation were TLR-expression dependent in vitro (*Figure 3D*) and no external antigen was provided to the cultures, TLR-signaling may directly link to BCR-signaling. Alternatively, TLR-signaling may 'license' subsequent self-antigen recognition, by altering the BCR-signaling complex. In support of the latter, we noted a strong increase in the frequency of PTC-binding B-1 cells during culture (*Figure 6D*), which may be due to the expansion of CpG-activated B-1 cell in response to PTC-antigen present on dead and dying cells in the cultures.

## Initial stimulation through IgM-BCR suppresses subsequent activation of B-1 cells via TLR- stimulation

We further analysed the impact of stimulation through the TLR or IgM-BCR on stimulation of B-1 cells by the other receptor. For that a set of in vitro experiments was conducted, in which FACS-purified, eFluor 670-stained and rested CD5+ body cavity B-1 cells and splenic follicular B-2 cells were stimulated in vitro with anti-IgM and/or CpG for a total of 72 hr at which time proliferation was assessed by flow cytometry. Stimulation with either anti-IgM or CpG for 2 hr followed by wash-out and then stimulation with the other stimulus for 70 hr showed no difference to stimulation with the second stimulus alone (not shown). However, initial stimulation for 24 hr followed by wash-out and restimulation with the second stimulus showed significant effects of initial IgM-BCR stimulation on subsequent CpG responsiveness by B-1 and B-2 cells (*Figure 7*). While B-1 cell responses to anti-IgM alone for 24 or 72 hr did not result in significant proliferation, when CpG was given after 24 hr

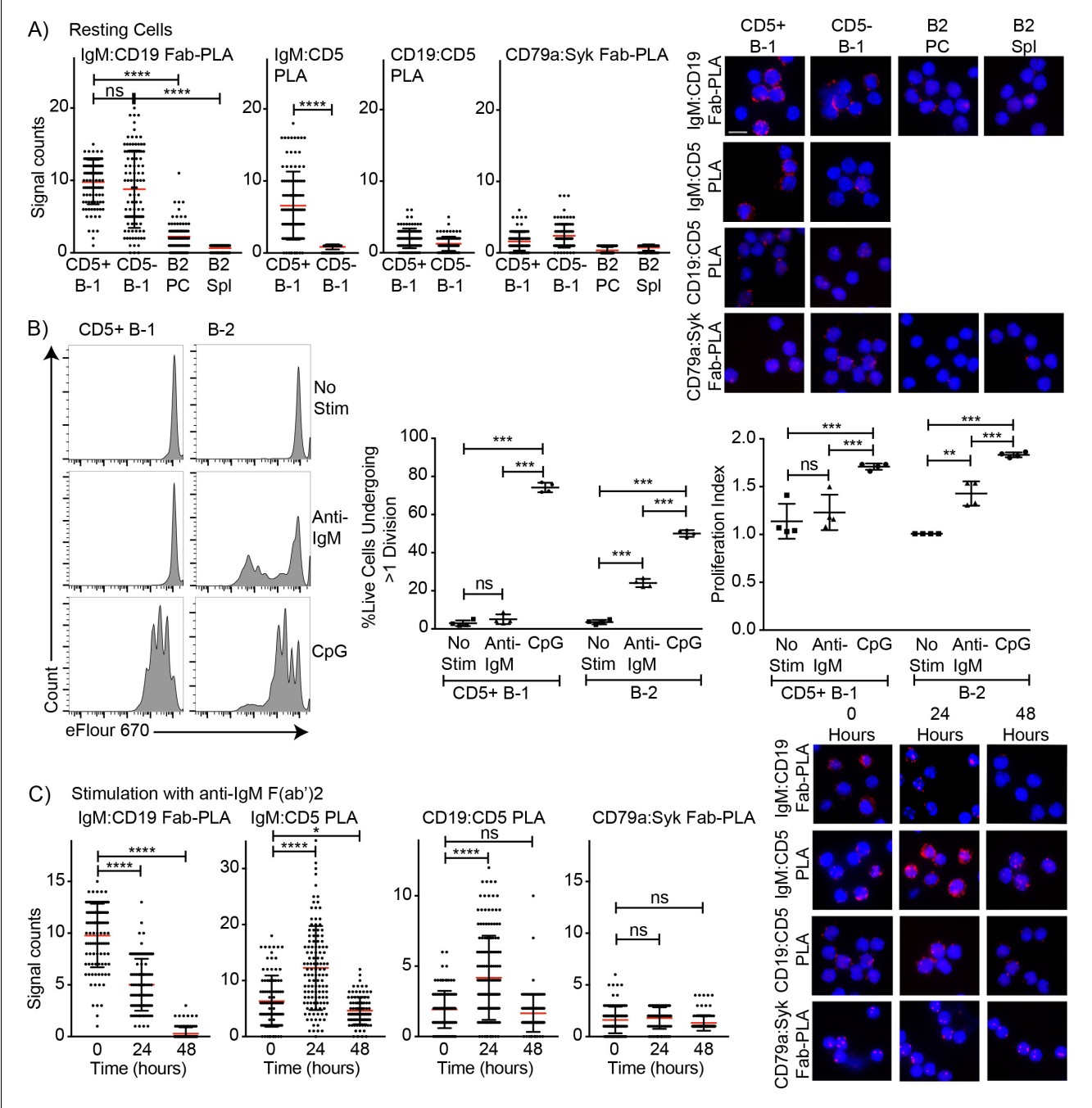

**Figure 5.** Association of CD5 with IgM-BCR in resting B-1a cells is increased after BCR-stimulation. (**A**) Indicated FACS-purified B cell subsets from the peritoneal cavity (PC) and spleen (Spl) of BALB/C mice were analyzed by proximal ligation assay for the following interactions (left to right): IgM:CD19, IgM:CD5, CD19:CD5 and CD79:syk. Left panel shows summarizes data on signal counts for 200 individual cells analyzed. Each symbol represents one cell, horizontal line indicates mean signal count per cell. Right panel show representative fluorescent images. (**B**) FACS-purified CD19hi CD23- CD5 + CD43+ B-1 cells from the peritoneal cavity and CD19+ CD23+ splenic B-2 cells of BALB/C mice were labeled with efluor670 and then cultured in the absence (top) or presence of 20 ug/ml anti-IgM (middle) or 10 ug/ml CpGs for 72 hr. Left panels show representative histogram plots, middle panel shows the % cells in each culture having undergone at least one cell division and right panel indicates the proliferation index (average number of proliferations undergone per divided cell). (**C**) Summary of proximal ligation assay results of B-1 cells purified as in (**A**) and then stimulated for indicated times with anti-IgM(Fab)₂. Interactions of the following proteins were analyzed on 200 cells per condition (left to right): IgM:CD19, IgM:CD5, CD19:CD5 and CD79:syk. Right panels shows representative fluorescent images from one experiment of at least two done. Values were compared using an unpaired Student's t test (*=p < 0.05, **=p < 0.005, ***=p < 0.0005, ****=p < 0.00005).
DOI: https://doi.org/10.7554/eLife.46997.007

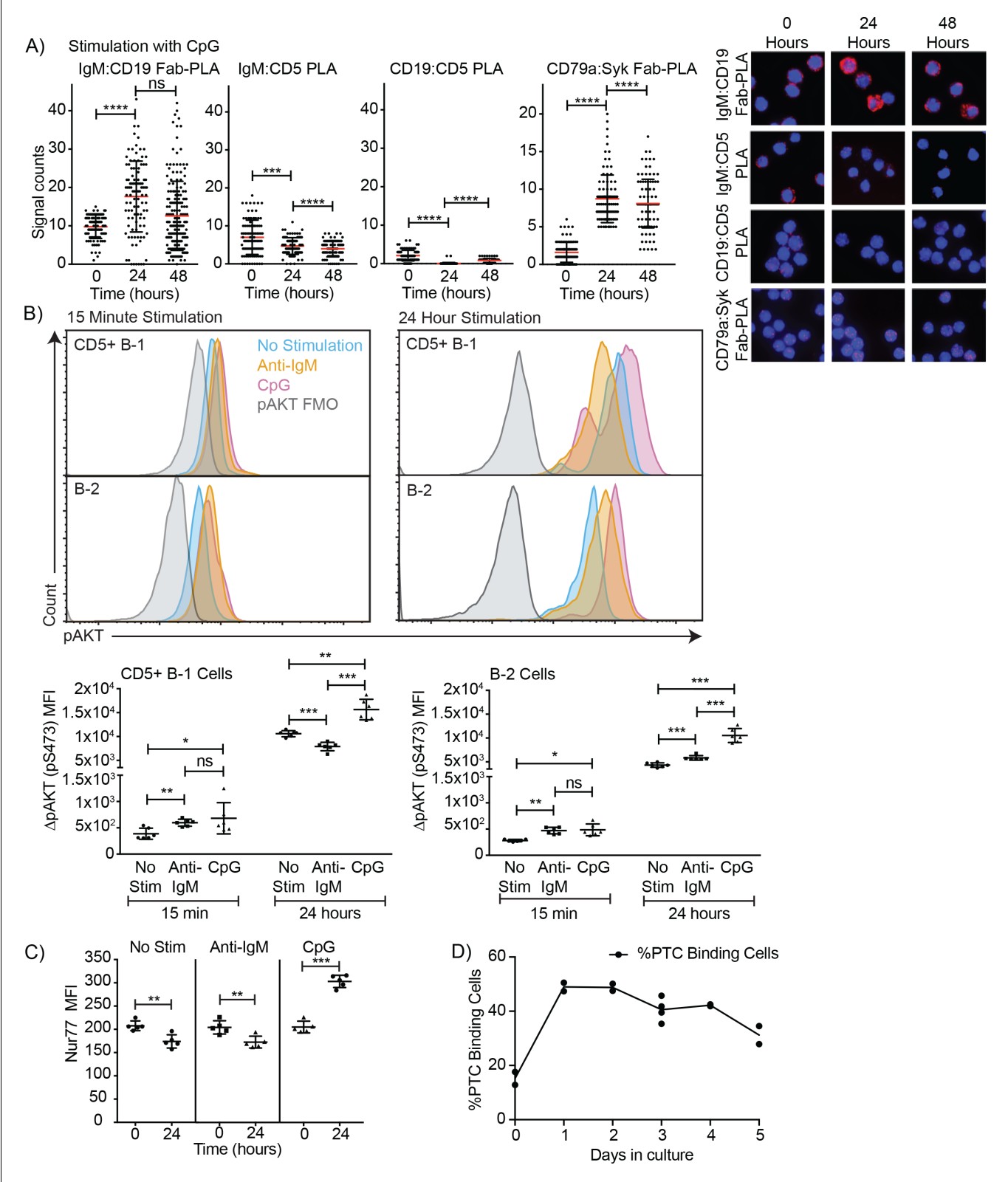

**Figure 6.** TLR-mediated stimulation of CD5+ B-1 cells alters the BCR-signalosome. (**A**) FACS-purified peritoneal cavity CD19$^{hi}$ CD23- CD43+ CD5+B-1 and splenic CD19+ CD23+ CD43 CD5- B-2 cell of BALB/C mice were stimulated for the indicated times with TLR9-agonist ODN7909 prior to analysis by proximal ligation assay, probing for the following interactions (left to right): IgM:CD19, IgM:CD5, CD19:CD5 and CD79:syk. Left panel summarizes data on signal counts for 200 individual cells analyzed. Each symbol represents one cell, horizontal line indicates mean signal count per cell. Right panel

*Figure 6 continued on next page*

*Figure 6 continued*

show representative fluorescent images. (**B**) Analysis of the phosphorylation status of Akt by probing for Akt pS473 by flow cytometry on FACS-purified CD19hi CD23- CD5+ CD43+ B-1 cells from the peritoneal cavity and CD19+ CD23+splenic B-2 cells of BALB/C mice. Top panels show representative histogram plots, bottom summarizes the results. (**C**) Mean fluorescence intensity ± SD of staining for the immediate early activation factor Nur77, in CD5+ B-1 cells isolated as described in (**A**) and cultured for up to 2 days in the absence and presence of the indicated stimuli. (**D**) Shown are % frequencies of live PtC-binding B-1 cells among live FACS-purified CD5+ peritoneal cavity B-1 cells cultured with LPS stimulation for the indicated times, as assessed by flow cytometry. Each symbol represents results obtained from one culture well. Results are representative from experiments conducted at least twice with multiple repeats done per experiment (n = 2–5). Results in D are combined from two independent experiments. Values were compared using an unpaired Student's t test (*=p < 0.05, **=p < 0.005, ***=p < 0.0005).

DOI: https://doi.org/10.7554/eLife.46997.009

anti-IgM stimulation a small but significant increase in proliferation was noted (***Figure 7A***). However, proliferation rates were greatly lower compared to stimulation with CpG first, followed by anti-IgM stimulation, or stimulation with CpG alone (***Figure 7A***). In contrast, initial anti-IgM stimulation of B-2 cells followed by stimulation with CpG resulted in the most robust B-2 cell proliferative response (***Figure 7B***), exceeding that of CpG stimulation alone or CpG stimulation followed by anti-IgM.

Thus B-1 and B-2 cells greatly differ in their responses not only to BCR but also to TLR stimulation. While TLR stimulation by B-2 cells induced a strong and synergistic enhancement of proliferation initiated by BCR-signaling, B-1 cells did not respond to BCR-mediated stimulation with proliferation, independent of whether the BCR signal was given first or after a TLR-stimulus. However, only when TLR stimulation was provided first, did B-1 cells show robust proliferation in context of anti-IgM BCR-signaling.

## Local IgM production following influenza infection depends on TLR expression

The data suggest that TLR-mediated stimulation alters the BCR-signalosome complex, which may drive B-1 cell responses to pathogens in vivo. Indeed, complete TLR-deficient mice (due to a lack of TLR2, TLR4 and Unc93) showed significant deficits in CD5+ B-1 cell responses following influenza infection (***Figure 8***). Significant increases in the ratios of CD5+ over CD5- and CD19+ CD43+ B cells were noted (***Figure 8A***). This suggested that CD5+ B-1 cells in TLR-deficient mice could enter the MedLN, consistent with our previous findings that this step is TLR-independent but Type I IFN-dependent (***Waffarn et al., 2015***), but once in the MedLN they were not activated via TLR-dependent signals, that is failed to downregulate CD5. Of importance, the lack of TLR-stimulation also resulted in a near complete loss of CD19$^{lo/-}$ IgM+ CD138+ B-1PC in the MedLN at day 5 of infection (***Figure 8A/B***) and a corresponding drop in IgM ASC in TLR-deficient compared to control mice at that timepoint (***Figure 8C***), while viral loads were similarly low in the lungs of both mouse strains (not shown). Generation of Ig-allotype chimeras in which only B-1 cells lacked TLR expression confirmed a B-1 cell-intrinsic requirement for TLR-signaling in B-1 cell differentiation to CD138+ ASC after influenza virus infection (***Figure 8D–F***).

Together the data demonstrate a linkage of TLR and BCR-signaling during B-1 cell responses to infections, with intrinsic TLR-mediated signaling triggering a rapid reorganization of the IgM-BCR-signalosome complex, including the removal of the BCR-signaling inhibitor CD5 and increased association of IgM-BCR with the co-receptor CD19, and the TLR-dependent differentiation of CD5+ B-1 to CD5- IgM-secreting B-1 and B-1PC.

## Discussion

Self-reactive, fetal and neonatal-developing B-1 cells do not respond to BCR-stimulation with clonal expansion. This was shown previously to be associated with their expression of the BCR-signaling inhibitor CD5, and a lack of fully functional CD19 signaling. It is consistent with the notion that these self-reactive cells must be silenced in order to avert the risk of autoimmune disease induction. Yet B-1 cells do respond rapidly to various infections with migration to secondary lymphoid tissues and with differentiation to IgM secreting cells. The present study resolves this conundrum by providing a mechanism by which B-1 cell can overcome their inherent BCR-signaling block, namely the TLR-mediated activation and reorganization of the BCR-signalosome complex. This non-redundant signal

induced the dissociation of CD5 from the IgM-BCR and eventual its loss from the cell surface of the initially CD5-expressing B-1 cells. It also enhanced the association between IgM and the co-stimulatory molecule CD19, and caused strong increases in CD79:Syk interaction and phosphorylation of BCR downstream effectors.

The study also clarifies the respective roles of the CD5+ and CD5- B-1 cells, previously suggested to form two distinct subsets 'B-1a' and 'B-1b'. By demonstrating that CD5+ B-1 cells respond to infections with both, influenza and *S. typhimurium*, the latter previously identified as an exclusive 'B-1b' response (*Gil-Cruz et al., 2009*; *Marshall et al., 2012*), with rapid downregulation of CD5, the study suggests that pathogen-induced responses by B-1 cells represent responses of CD5+ B-1 cells (*Haas et al., 2005*; *Alugupalli et al., 2003*; *Alugupalli et al., 2004*; *Gil-Cruz et al., 2009*; *Marshall et al., 2012*). The lack of CD5-expression on B-1 cells may thus more broadly identify activated and differentiated B-1 cells, a conclusion supported also by the work in the accompanying manuscript by *Kreuk et al. (2019)*. This is consistent with previous reports on the phenotype of natural IgM-secreting cells as mostly CD5- (*Savage et al., 2017*; *Ohdan et al., 2000*), and could explain earlier reports that CD5+, not CD5- B-1 cells form natural IgM secreting cells (*Haas et al., 2005*; *Masmoudi et al., 1990*; *Hayakawa et al., 1984*). In addition, it could explain the findings that CD5- B-1 cells contain CD5- memory-like B-1 cells in the body cavities of previously infected mice (*Alugupalli et al., 2004*; *Foote and Kearney, 2009*) which also likely respond to antigen.

Thus, the data presented here and in the accompanying work by Kreuk and colleagues are inconsistent with models that consider B-1 cell responses as a division of labor between two subsets of B-1 cells: B-1a and B-1b cells, where CD5+ B-1a contribute 'natural' IgM and CD5- B-1b the induced IgM, proposed previously (*Haas et al., 2005*; *Alugupalli and Gerstein, 2005*). Given CD5 expression is dynamically expressed and thus cannot be used to identify B-1 cell subsets, and no other clearly subset-defining differences have been reported to-date, we suggest that the separation of B-1 cells into B-1a and B-1b 'sister populations' be revoked, and that instead these cells are simply referred to as B-1 cells. If distinctions in CD5 expression are important, those could be indicated by describing B-1 cells as CD5+/CD5- instead.

Our data do not exclude the possibility that some B-1 cells develop which either express no, or undetectable levels of surface CD5, as described previously (*Kantor et al., 1992*). Given the known functions of CD5 as an inhibitor of BCR-signaling (*Hippen et al., 2000*; *Punt et al., 1994*; *Azzam et al., 1998*; *Perez-Villar et al., 1999*) and the fact that CD5-expression levels on thymocytes correlated with the strengths of the positively selecting TCR–MHC-ligand interactions (*Azzam et al., 1998*), such CD5$^{lo/neg}$ B-1 cells might have lower levels of self-reactivity (*Hayakawa et al., 1999*; *Lalor and Morahan, 1990*; *Mercolino et al., 1988*; *Yang et al., 2015*) and lack the need for CD5-mediated silencing of BCR-signaling in order to avoid inappropriate hyperactivation of these self-reactive B cells. De novo development of both CD5+ and CD5- B-1 cells has been reported to occur also in stromal cell cultures seeded with B-1 cell precursors (*Montecino-Rodriguez et al., 2006*). It remains to be explored whether the presence of DAMPS in the in vitro cultures could contribute to the loss of CD5 on otherwise CD5+ B-1 cells, or whether these cells never expressed CD5. Our data are not consistent with early reports suggesting that CD5+ B-1 cells could only reconstitute themselves, but not CD5- B-1 cells (*Kantor et al., 1992*; *Stall et al., 1992*), as reconstitution of neonatal mice with even very highly FACS-purified body cavity CD5+ B-1 cells led to significant numbers of CD5- B-1 cells recovered from these mice (*Choi and Baumgarth, 2008* and *Figure 4—figure supplement 1*).

The mechanisms by which the TLR-induced reorganization of the BCR-signalosome drives the differentiation of B-1 cells to IgM-secreting B-1PC will require future in-depth analysis. We favor the concept of a 'licensing' step, in which initial TLR-stimulation supports subsequent BCR-stimulation through the here observed reorganization of the BCR-signalosome. This is supported also by the in vitro studies which demonstrated that an initial engagement of the TLR could overcome the block in B-1 cell proliferation after BCR stimulation, while an initial BCR stimulus followed by TLR stimulation failed to induce B-1 cell proliferation (*Figure 7*). Such a licensing step may explain the recent data by Kreuk and colleagues (see *Kreuk et al., 2019*), which showed a degree of B-1 response specificity, and thus presumably BCR-responsiveness, that was dependent on the type of TLR stimulus provided to the B-1 cell. However, how such licensing step could induce antigen-specific B-1 cell responses that are dependent on the specificity of the TLR remains to be fully elucidated.

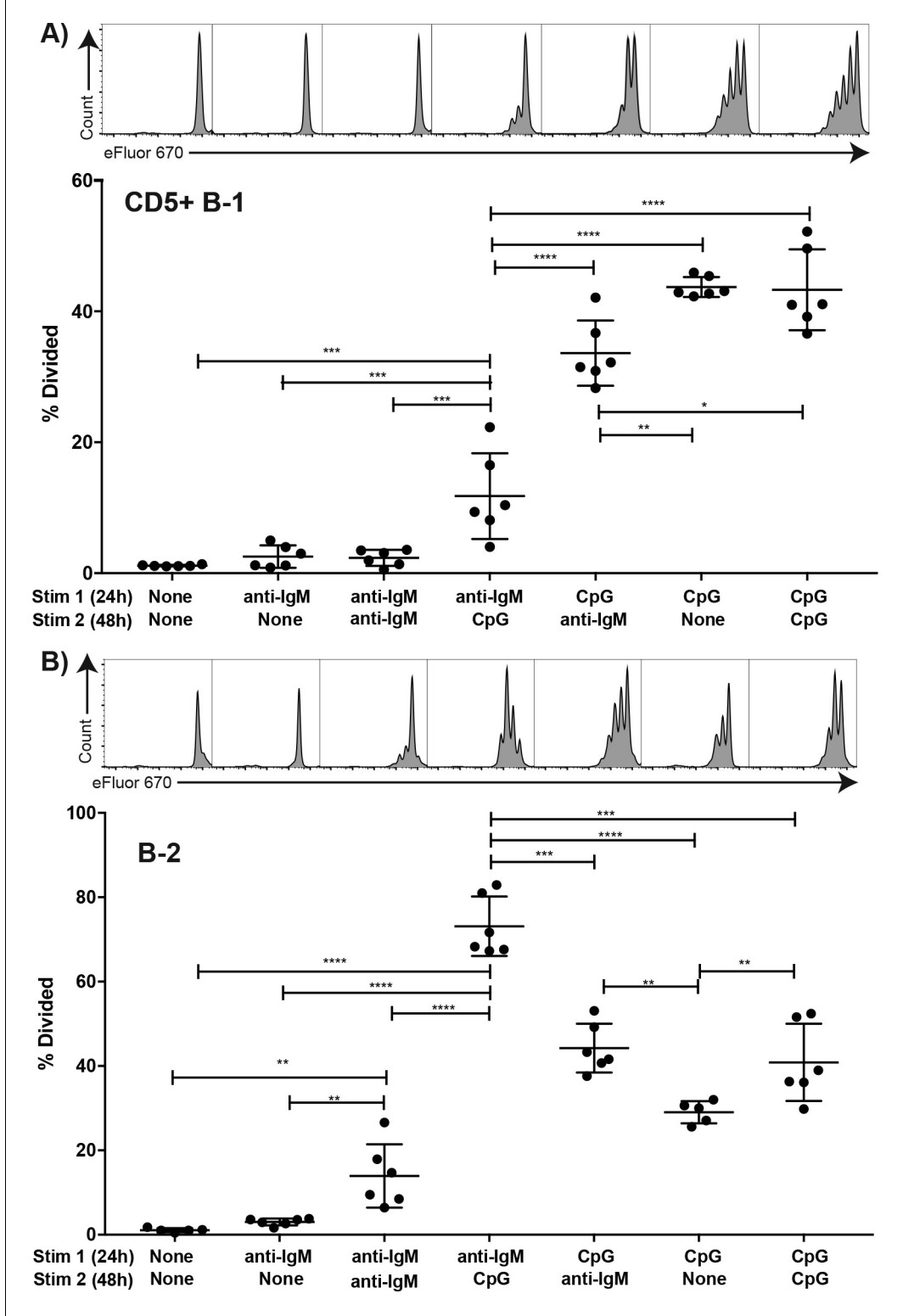

**Figure 7.** TLR but not BCR-stimulation induces CD5+ B-1 cell proliferation FACS-purified. (**A**) peritoneal cavity CD19hiCD23- CD43+ CD5+ B-1 and (**B**) splenic CD19+ CD23+ CD43 CD5- B-2 cell from BALB/C mice were labeled with eFluor 670, rested for 2 hr and then cultured for 24 hr with the indicated stimulus 1 (none/anti-IgM at 10 μg/ml or CpG ODN7909 at 5 μg/ml), washed and recultured for 48 hr with stimulus 2 (none/anti-IgM at 10 μg/ ml/ CpG ODN7909 at 5 μg/ml) prior to analysis for efluor 670 staining. Top panels show representative FACS histogram plots and bottom panels shows the % cells in each culture having undergone at least one cell division.
*Figure 7 continued on next page*

*Figure 7 continued*

Each symbol represents results from one culture well, horizontal line indicates mean for the group. Results are compiled from two independent experiments. Statistical analysis was done by one-way ANOVA, followed by an unpaired Student's t test with Holm-Sidak correction for multiple comparisons (*p<0.05, **p<0.005, ***p<0.0005, ****p<0.00005).

DOI: https://doi.org/10.7554/eLife.46997.010

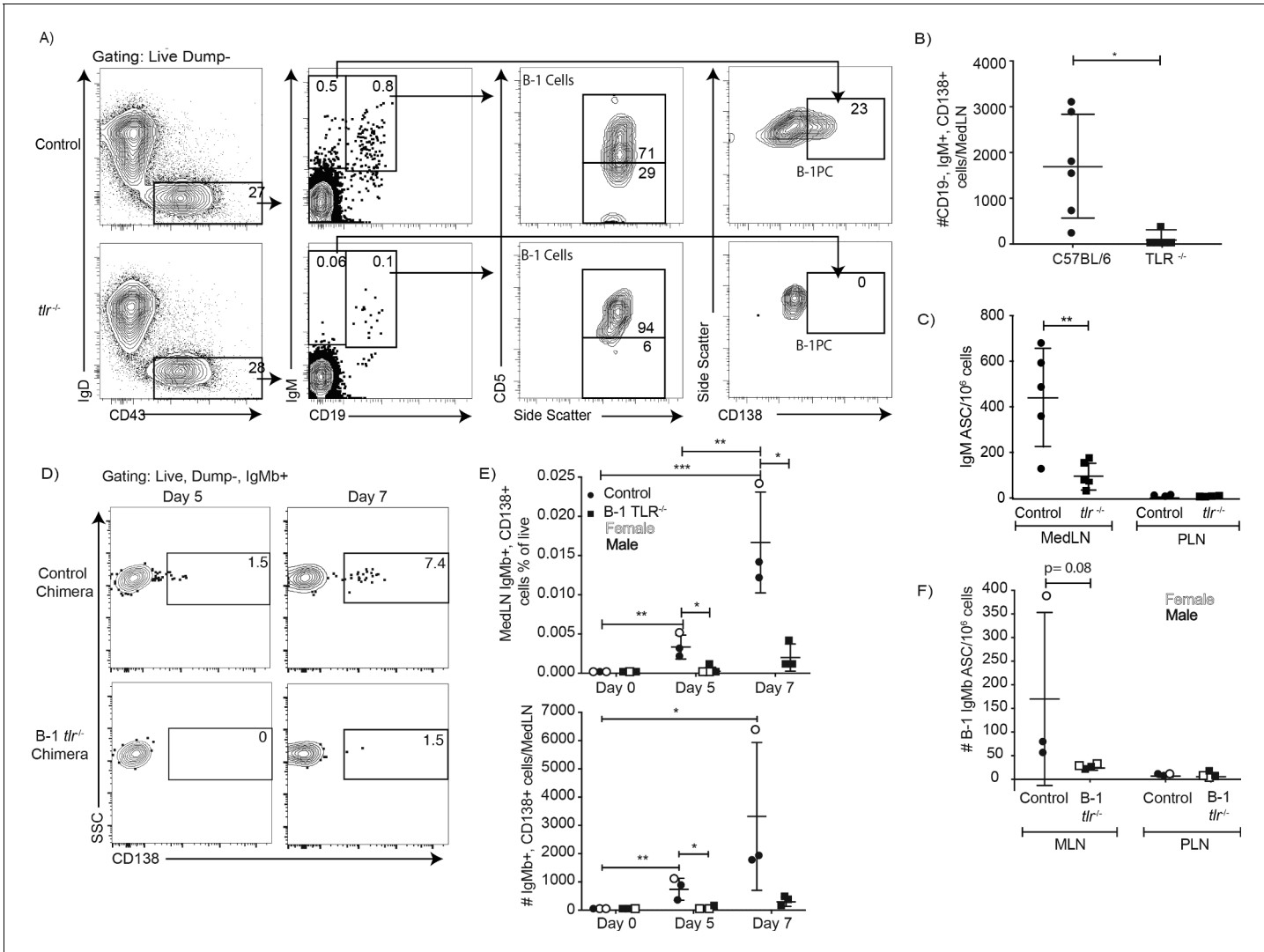

**Figure 8.** TLR-mediated stimulation is required for maximal IgM responses to influenza virus infection. (**A**) C57BL/6 (n = 5) and congenic total TLR-deficient mice (n = 5; lacking TLR2, TLR4 and Unc93) were infected with influenza A/Puerto Rico/8/34 for 5 days. Shown are representative FACS plots from control C57BL/6 (top) and TLR-deficient (bottom) mice FACS analysis of MedLN for the presence of B-1 and B-1PC. (**B**) Number of B-1PC per MedLN as assessed by FACS and (**C**) number of IgM-secreting cells in MedLN as assessed by ELISPOT analysis. (**D-F**) Similar analysis as for A-C but using allotype chimeras generated with wild type recipients and B-1 cells from either C57BL/6 or *tlr-/-* mice. (**D**) Representative FACS analysis of CD138 + B-1PC pre-gated for live, dump-, B-1 donor (IgMb+) cells in MedLN on days 5 and 7 after influenza infection. (**E**) Mean ± SD of data summarized from analysis shown in D. (**F**) Mean ± SD of B-1 IgM-ASC in MedLN on day 7 after infection, as assessed by ELISPOT. Each symbol represents results from one mouse with female mice shown as open symbols, males as closed symbols. Results are combined from two independent experiments. Values were compared using an unpaired Student's t test (*=p < 0.05, **=p < 0.005, n.s. not significant).

DOI: https://doi.org/10.7554/eLife.46997.011

An attractive alternative and perhaps simpler concept would be that the BCR-mediated binding to antigen and then its uptake by BCR-internalization leads to engagement of particular TLR, which then synergize with BCR-mediated stimulation and support the strong proliferation and differentiation of antigen-specific B cells. This concept is very consistent with the outcome of B-2 cell stimulation in vitro (*Figure 7*), where TLR stimulation strongly and synergistically enhanced the initial BCR-mediated activation of these cells. However, the model cannot explain the complete lack of B-1 cell proliferation in response to BCR-engagement, the strong inhibition of TLR-mediated proliferation when preceded by BCR-engagement, and the BCR-signaling induced enhanced unresponsiveness of the BCR signalosome at the molecular level, as demonstrated by a failure to enhance CD79:syk interaction and upregulation of pAkt.

A potential third model would involve a direct linkage of TLR and BCR-signalosome effects. For example, low-affinity BCR-antigen interactions might be enabled by having DAMPS and PAMPS first bind to TLR on the B cell surface, which then brings these antigens in close proximity to the BCR, triggering antigen-specific BCR activation event. Although in some cultures we did not provide antigens other than the TLR-ligands to the in vitro cultures, dead and dying cells provide ample antigens, including PTC, that could have stimulated these cells. This would potentially explain why the stimulation of B-1 cells with TLR-agonists activates BCR signaling pathways, and why we found such strong enrichment for PTC-binders among cultures of TLR-stimulated B-1 cells. However, not all TLR are expressed on the cell surface, and their initial engagement prior to that of the BCR would necessitate the initial BCR-independent phagocytosis of antigen by B-1 cells, which others have reported (*Popi et al., 2016*).

The lack of phenotypic differences between CD5- B-1 cells and B-2 cell-derived non-switched plasmablasts, both expressing CD19, CD43, low levels of CD45 and lack IgD, further complicates the identification of responding B cell subsets in vivo, as demonstrated with the analysis to the *Salmonella* antigen OmpD. Using allotype-marked B cell lineages, we show here that anti-OmpD secreting B cells were derived predominantly from B-2 cells, and not as previously suggested from B-1b cells (*Gil-Cruz et al., 2009*). When CD5+ B-1 cells lose CD5, some also seem to begin to express CD138, thus becoming indistinguishable from B-2-derived plasma cells and plasmablasts that carry the same phenotype. Some investigators used expression of CD11b to identify B-1 versus B-2 cells in infectious models (*Haas et al., 2005*; *Gil-Cruz et al., 2009*), but this marker is also dynamically regulated depending on tissue site and B-1 cell activation status (*Waffarn et al., 2015*; *Ohdan et al., 2000*; *Yang et al., 2007*). Recent lineage-tracing approaches (*Yuan et al., 2012*; *Zhou et al., 2015*; *Montecino-Rodriguez et al., 2016*) may help the development of novel approaches or markers that can unequivocally identify B-1 cells. In the meantime, the use of neonatal B-1 allotype-chimeras remains a valuable tool for such analyzes.

Taken together, our data suggest that the BCR-complex composition on neonatally-derived, self-reactive B-1 cells is controlled by TLR-mediated signals, preventing inappropriate activation and autoimmune disease on the one hand, while facilitating rapid B-1 cell participation in anti-viral and anti-bacterial infections on the other. TLR-signaling thereby influences not only innate B-1 cell activation, but may also affect their antigen-specific responses.

# Materials and methods

**Key resources table**

| Reagent type (species) or resource | Designation | Source or reference | Identifiers | Additional information |
|---|---|---|---|---|
| Strain, strain background (mouse) musculus (mouse), C57BL/6J (M + F) | C57BL/6, Control, Ighb | The Jackson Laboratories | Stock 000664 | *Mus musculus* (mouse), C57BL/6J (M + F) |
| Strain, strain background (mouse) | Igha | The Jackson Laboratories | Stock 001317 | Mouse, B6.Cg-Gpi1aThy1aIgha/J (M + F) |

*Continued on next page*

*Continued*

| Reagent type (species) or resource | Designation | Source or reference | Identifiers | Additional information |
|---|---|---|---|---|
| Strain, strain background (mouse) | BALB/C | The Jackson Laboratories | Stock 000651 | Mouse, BALB/CJ mice (F) |
| Strain, strain background (mouse) | Blimp-1YFP | *Rutishauser et al., 2009* | | Breeding pairs from Michel Nussenzweig (The Rockefeller University) Mouse, B6-Cg-Tg(PRDM1- EYFP)(1Mnz) (M + F) |
| Strain, strain background (mouse) | TLR-/- | Other | | Breeding pairs from Greg Barton (The University of California, Berkeley) Mouse, Tlr2-/- Tlr4-/- Unc93b1(3d/3d) (M + F) |
| Strain, strain background (mouse) | Chimera | *Lalor et al., 1989* | | Generated in-house Mouse, Igha/Ighb B-1 Cell Neonatal Chimera |
| Strain, strain background (mouse) | Chimera | *Lalor et al., 1989* | | Generated in-house Mouse, Igha/Ighb-YFP B-1 Cell Neonatal Chimera |

## Mice

8-16 week old male and female C57BL/6J or female BALB/c mice and breeding pairs of B6.Cg-$Gpi1^aThy1^aIgh^a$/J ($Igh^a$) mice were purchased from The Jackson Laboratory. Female, 10 weeks old BALB/C mice were purchased from Jackson Laboratory. B6-Cg-Tg($prdmi$-$EYFP$)$^{1Mnz}$ (Blimp-1 YFP) breeders were kindly provided by Michel Nussenzweig (The Rockefeller University, NY) and breeding pairs of $Tlr2^{-/-}$ x $Tlr4^{-/-}$ x $Unc93b1^{3d/3d}$ (TLR-deficient) mice by Greg Barton (University of California, Berkeley, CA). Mice were housed under SPF conditions in micro-isolator cages with food and water provided ad libitum. Mice were euthanized by overexposure to carbon dioxide. All procedures were approved by the UC Davis Animal Care and Use Committee.

## Chimera generation

Neonatal chimeric mice were generated as described previously (*Lalor et al., 1989*; *Baumgarth et al., 2000*; *Baumgarth et al., 1999*). Briefly, one-day old $Igh^a$ C57BL/6 congenic mice were injected intraperitoneally with anti-IgM$^a$ (DS-1.1) diluted in PBS. On day 2 or three after birth mice were injected with total peritoneal cavity wash out, or with FACS-purified dump- CD19+ CD43 + CD5+ and/or CD5- B-1 cells from C57BL/6 ($Igh^b$) mice. Over the next 6 weeks, these donor B-1 cells expand to fill all tested B-1 compartments, including the peritoneal and pleural body cavities, bone marrow, spleen, lymph nodes (mediastinal, mesenteric, inguinal, axilaris, cervical), gastrointestinal tract and lung, while host B-1 cells are depleted. Intraperitoneal anti-IgM$^a$ injections were continued twice weekly until mice reached 6 weeks of age. Mice were then rested for at least 6 weeks before use, for reconstitution of the conventional B cell populations from the host bone marrow. Due to the lack of significant B-1 cell development from bone marrow precursors after 6 weeks (Dorshkind et al.), the reconstituted mouse has exclusively host-derived B-2 cells ($Igh^a$) as well as a B-1 cell compartment that is 80–95% donor-derived ($Igh^b$).

## Influenza virus infection

Influenza A/Puerto Rico/8/34 was grown and harvested as previously described (*Doucett et al., 2005*). Mice were anesthetized with isoflurane and virus was diluted to a previously titrated sublethal dose of infection and administered intranasally in PBS.

## Salmonella typhimurium infection

Oral infections with *S. typhimurium* were performed following previously described protocols (*O'Donnell et al., 2015*). *S. typhimurium*, strain SL1344, kindly provided by Stephen McSorley (University of California, Davis, CA), was grown overnight at 37°C in Luria-Bertani broth. A known volume of bacteria were centrifuged for 20 min at 6,000–8,000 rcf at 4°C after concentration was determined by spectrophotometer reading at $OD_{600}$. Bacterial pellets were resuspended in mouse drinking water to a concentration of $10^9$ CFU/ml. Water was provided to mice ad lib.

## Flow cytometry and sorting

Tissues were processed and stained as described previously (*Rothaeusler and Baumgarth, 2006*). Briefly, single cell suspensions of spleen, lymph node, and Peyer's patches were obtained by grinding tissues between the frosted ends of two microscope slides, then resuspended in 'Staining Media' (*Rothaeusler and Baumgarth, 2006*). Peritoneal cavity washout was obtained by introducing Staining Media into the peritoneal cavity with a glass pipet and bulb, agitating the abdomen, and then removing the media. Samples were filtered through nylon mesh and treated with ACK lysis buffer as needed. Cell counts were performed using Trypan Blue exclusion to identify live cells.

Fc receptors were blocked using anti-CD16/32 antibody (2.4G2) and cells were stained using fluorochrome conjugates generated in-house unless otherwise specified against the following antigens: CD19 (clone ID3)-Cy5PE, allophycocyanin, FITC, CD4- (GK1.5), CD8a- (53–6.7), CD90.2- (30H12.1), Gr1- (RB68-C5), F4/80- (F4/80), and NK1.1- (PK136) Pacific blue ('Dump'), CD43- (S7) allophycocyanin or PE, IgM- (331) allophycocyanin, Cy7-allophycocyanin, FITC, Alexa700, IgM[a]- (DS-1.1) allophycocyanin, biotin, IgM[b]- (AF6-78.2.5) allophycocyanin, FITC, biotin, CD5- (53–7.8) PE, biotin, IgD- (11-26) Cy7PE, Cy5.5PE, CD138- (281-2) allophycocyanin, PE; CD138-BV605 (BD Bioscience), CD19-BV786, PE-CF594 (BD Bioscience), SA-Qdot605 (Invitrogen), SA- allophycocyanin (eBioscience), BrDU-FITC (BD Bioscience), B220 (CD45R) APC-eFluor 780 (Invitrogen) and CD23-biotin (eBioscience), BV605, BV711 (BD Bioscience). PTC-FITC liposomes were a kind gift of Aaron Kantor (Stanford University, CA). Dead cells were identified using Live/Dead Fixable Aqua or Live/Dead Fixable Violet stain (Invitrogen).

Intracellular staining: Cells were surfaced stained, then fixed (eBiosience IC Fixation Buffer) for 30 min and then permeabilized (eBioscience Permeablization Buffer) for 30 min, followed by staining with anti-Nur77-Alexa Fluor 488 for 30 min all at room temperature.

Phosphoflow: Cells were fixed (BD Cytofix) for 12 min at 37°C. Cells were then permeabilized (BD Perm Buffer III) for 30 min on ice and intracellularly stained with anti-phospho-Akt-Alexa Fluor 488 for 30 min on ice.

FACS analysis was done using either a 4-laser, 22-parameter LSR Fortessa (BD Bioscience) or a 3-laser FACSAria (BD Bioscience). Cells were sorted as previously described (*Rothaeusler and Baumgarth, 2006*) using the FACSAria and a 100 µm nozzle. Data were analyzed using FlowJo software (FlowJo LLC, kind gift of Adam Treister).

## Elisa

Sandwich ELISA was performed as previously described (*Rothaeusler and Baumgarth, 2006*). Briefly, MaxiSorp 96 well plates (ThermoFisher) were coated with anti-IgM (Southern Biotech) and nonspecific binding was blocked with 1% NCS/0.1% dried milk powder, 0.05% Tween20 in PBS ('ELISA Blocking Buffer'). Two-fold serial dilutions in PBS of culture supernatants and an IgM standard (Southern Biotech) were added to the plates at previously optimized starting dilutions. Binding was revealed with biotinylated anti-IgM (Southern Biotech), Streptavidin-Horseradish Peroxidase, both diluted in ELISA Blocking Buffer, and 0.005% 3,3',5,5'-tetramethylbenzidine (TMB)/0.015% hydrogen peroxide in 0.05 M citric acid. The reaction was stopped with 1N sulfuric acid. Antibody concentrations were determined by measuring sample absorbance on a spectrophotometer (SpectraMax M5, Molecular Devices) at 450 nm (595 nm reference wavelength) and then compared to a standard curve created with a mouse IgM standard (Southern Biotech) of known concentration.

## Culture and proliferation dye labeling

After FACS sorting, cells were labeled with Efluor670 or CFSE at previously determined optimal concentrations, by incubation at 37°C for 10 mins., then washed three times with staining medium

containing 10% neonatal calf serum and resuspended into 'Culture Media' (RPMI 1640 with 10% heat inactivated fetal bovine serum, 292 µg/ml L-glutamine, 100 Units/ml penicillin, 100 µg/ml strep-tomycin, and 50 µM 2-mercaptoethanol). Cells were plated at $10^5$ cells/well of 96-well U bottom tissue culture plates (BD Bioscience), and unless otherwise indicated, cultured at 37°C/5% $CO_2$ for 3 days. When indicated, LPS at 10 µg/ml, Mycobacterium TB lipids at 20 µg/ml (BIA), Imiquimod (R837, InvivoGen) at 1 µg/ml, CpG ODN 7909 at 5 µg/ml or anti-IgM (Fab)$_2$ at 10–20 ug/ml were added to the wells. Cell enumeration after culture was performed using Molecular Probes Count-Bright Beads (Thermo Fisher) by flow cytometry, per manufacturer instructions. After culture, culture plates were spun and supernatant was collected and stored at −20°C, and cells were stained for FACS.

## Elispot

IgM antibody secreting cells were enumerated as previously described (*Doucett et al., 2005*). Briefly, 96 well ELISPOT plates (Multi-Screen HA Filtration, Millipore) were coated overnight with anti-IgM (331) or recombinant OmpD (MyBioSource) and non-specific binding was blocked with 4% Bovine Serum Albumin (BSA)/PBS. Cell suspensions were processed, counted, and directly plated in culture medium into ELISPOT wells and subsequently serially diluted two-fold, or they were FACS-sorted directly into culture media-containing ELISPOT wells. Cells were incubated overnight at 37°C/5% $CO_2$. Binding was revealed with biotinylated anti-IgM (Southern Biotech), anti-IgM[a] (BD Bioscience), or anti-IgM[b] (BD Bioscience), Streptavidin-Horseradish Peroxidase (Vector Labs) both diluted in 2% BSA/PBS, and 3.3 mg 3-amino-9-ethylcarbazole (Sigma Aldrich) dissolved in dimethyl formamide/0.015% hydrogen perioxide/0.1M sodium acetate. The reaction was stopped with water. Spots were enumerated using the AID EliSpot Reader System (Autoimmun Diagnostika, Strassberg, Germany).

## qRT-PCR

mRNA was isolated from cells using the RNeasy mini kit (Qiagen), per manufacturer instructions. cDNA was generated using random hexamer primers and SuperSript II reverse transcriptase (Invitrogen). qRT-PCR was performed using commercially available Taqman primer/probes for *cd5* and *ubc* (Thermo Fisher).

## BrDU labeling

Mice were injected intraperitoneally with 1 mg of BrDU (Sigma-Aldrich) per mouse diluted in 100 µL PBS, 24 hr before tissue collection. Staining for BrDU was performed as described previously (*Rothaeusler and Baumgarth, 2006*).

## Proximity Ligation Assay (PLA)

After FACS sorting, cells were resuspended in RPMI and rested for at least two hours before designated stimuli were added to culture media. Stimulated and unstimulated cells were cultured for 5 min, and 24 and 48 hr prior to PLA. PLA was performed as previously described (*Kläsener et al., 2014*). In brief: For PLA-probes against specific targets, the following unlabeled Abs were used: anti-IgM (Biolegend, clone RMM-1), anti-CD79a (Thermo Fisher, clone 24C2.5), anti-CD5 (Biolegend, clone 53–7.3), anti-Syk (Biolegend, clone Syk-01), and anti-CD19 (Biolegend, clone 6D5). Fab fragments against CD79a, Syk, IgM, and CD19 were prepared with Pierce Fab Micro preparation kit (Thermo Scientific) using immobilized papain according to the manufacturer's protocol. After desalting (Zeba spin desalting columns, Thermo Scientific), all antibodies were coupled with PLA Probe-maker Plus or Minus oligonucleotides (Sigma-Aldrich) to generate PLA-probes. For in situ PLA, B cells were settled on polytetrafluoroethylene slides (Thermo Fisher Scientific) for 30 min at 37°C. BCR. Cells were fixed with paraformaldehyde 4%, for 20 min. For intracellular PLA, B cells were permeabilized with 0,5% Saponin for 30 min at room temperature, and blocked for 30 min with Blocking buffer (containing 25 µg/ml sonicated salmon sperm DNA, and 250 µg/ml bovine serum albumin). PLA was performed with the Duolink In-Situ-Orange kit. Resulting samples were directly mounted on slides with DAPI Fluoromount-G (SouthernBiotech) to visualize the PLA signals in relationship to the nuclei. Microscope images were acquired with a Leica DMi8 microscope, 63 oil objective (Leica-microsystems). For each experiment a minimum of 100 B-1a/B-1b/B-2 peritoneal cavity or 1000

splenic B-2 cells from several images were analyzed with CellProfiler-3.0.0 (CellProfiler.org). Raw data were exported to Prism7 (GraphPad, La Jolla, CA). For each sample, the mean PLA signal count per cell was calculated from the corresponding images and the statistical significance with Mann–Whitney test.

## Statistical analysis

Statistical analysis was done using a two-tailed Student t test with help of Prism software (GraphPad Software). For time-course data, an ANOVA was performed with the help of Prism software, and if significant, Student t tests were performed to determine which time points were significant. When multiple comparisons were run on the same sets of data, Holm-Sidak correction was applied, using Prism software. $p < 0.05$ was considered statistically significant.

## Acknowledgements

We thank Drs. Joseph Benoun and Stephen McSorley (UC Davis) for expert help in conducting infections with *S typhimurium*, Dr. Aaron Kantor (Stanford University) for PtC liposomes, and Dr. Greg Barton (UC Berkeley) for TLR-deficient mice and discussions. This work was supported through NIH/NIAID grants U19-AI109962 (NB) and R01-AI117890 (NB), the National Center for Advancing Translational Sciences, NIH, through grant number UL1 TR000002 and linked award TL1 TR000133 (HPS), the NIH −2T32OD010931-09 (HPS), NIH – 5T35OD010956 (HPS) and the T-32 AI060555 (HPS) and NIH- T32 OD011147 (FLS), and through the Excellence Initiative of the German Federal and State Governments (EXC 294) and TRR130-P02 of the Deutsche Forschungsgemeinschaft (KK and MR).

## Additional information

### Funding

| Funder | Grant reference number | Author |
| --- | --- | --- |
| National Institute of Allergy and Infectious Diseases | U19-AI109962 | Nicole Baumgarth |
| National Institute of Allergy and Infectious Diseases | R01-AI117890 | Nicole Baumgarth |
| National Institutes of Health | T32 OD011147 | Fauna L Smith |
| National Institutes of Health | T32 AI060555 | Hannah P Savage |
| National Institutes of Health | 2T32OD010931-09 | Hannah P Savage |
| National Institutes of Health | 5T35OD010956-14 | Hannah P Savage |
| National Center for Advancing Translational Sciences | TL1 TR000133 | Hannah P Savage |
| The German Federal and State Governments Excellence Initiative | EXC 294 | Kathrin Kläsener Michael Reth |
| Deutsche Forschungsgemeinschaft | TRR130-P02 | Kathrin Kläsener Michael Reth |

The funders had no role in study design, data collection and interpretation, or the decision to submit the work for publication.

### Author contributions

Hannah P Savage, Conceptualization, Data curation, Formal analysis, Investigation, Methodology, Writing—original draft, Writing—review and editing; Kathrin Kläsener, Fauna L Smith, Data curation, Formal analysis, Investigation, Methodology, Writing—review and editing; Zheng Luo, Resources, Formal analysis, Validation; Michael Reth, Conceptualization, Methodology, Writing—review and editing; Nicole Baumgarth, Conceptualization, Resources, Formal analysis, Supervision, Funding acquisition, Investigation, Methodology, Writing—original draft, Writing—review and editing

## Author ORCIDs

Hannah P Savage [ID] https://orcid.org/0000-0002-1057-7239
Kathrin Kläsener [ID] http://orcid.org/0000-0002-5969-2553
Nicole Baumgarth [ID] https://orcid.org/0000-0002-2891-4483

## Ethics

Animal experimentation: This study was performed in strict accordance with the recommendations in the Guide for the Care and Use of Laboratory Animals of the National Institutes of Health. All procedures and experiments involving animals were approved by the Animal Use and Care Committee of University of California, Davis (protocol #20556).

## Decision letter and Author response

Decision letter https://doi.org/10.7554/eLife.46997.014
Author response https://doi.org/10.7554/eLife.46997.015

## Additional files

### Supplementary files

• Transparent reporting form
DOI: https://doi.org/10.7554/eLife.46997.012

### Data availability

All data generated or analyzed in this study are included in the manuscript and supporting files.

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
