## [Decision Letter]

Thank you for submitting your article "TLR-signaling induces reorganization of the IgM-BCR complex regulating B-1 cell responses to infections" for consideration by *eLife*. Your article has been reviewed by three peer reviewers, and the evaluation has been overseen by a Reviewing Editor and Wendy Garrett as the Senior Editor. The reviewers have opted to remain anonymous.

Thank you very much for your patience in the reviewing process. Since the paper was submitted with "B cell receptor and TLR signaling coordinate to control distinct B-1 responses to both self and the microbiota" by Dr Barton and colleagues, we aimed to secure reviewers (Nos. 1 and 2) for both papers, so that the complementary nature could be fairly considered. The reviewers have discussed the reviews with one another and the Reviewing Editor has drafted this decision to help you prepare a revised submission.

A key point in the discussion was the B cell-intrinsic nature of the TLR signaling in the other paper. We have requested a revision of both papers, and we would be grateful if you could liaise especially on this point to call out the complementarity.

Reviewer #1:

In this manuscript, Baumgarth et al. describe their discovery and characterization of a novel mechanism by which innate signals can license B1 cells to respond to antigen receptor signaling and produce IgM. Specifically, they show that B cell-intrinsic TLR signaling can trigger the downregulation of the inhibitor of BCR signaling CD5, which effectively blocks antigen recognition-based activation of CD5+ B-1 cells. In so doing, the authors resolve multiple long-standing mysteries surrounding the mechanisms by which CD5+ B1 cells respond to antigen-dependent stimuli, overturn previously held views regarding the cellular origins of specific IgM responses, and uncover a defined mechanism by which B cells integrate innate and adaptive stimuli. The experiments that support these main conclusions are both elegant and, in my opinion, performed to an exceptionally high standard-e.g., the extensive use of neonatal B-1 allotype-chimeras enabled detailed and definitive examinations of the roles of B cell subsets in vivo. Overall, I have little to criticize regarding the veracity of the work and believe that this will be an important paper. My one remaining confusion has to do with the order of engagement of the innate versus adaptive stimuli. The authors speculate a bit about this in the discussion, but as far as I can tell don't directly address this experimentally. To me, the logical order of events under in vivo conditions (in response to self-derived debris or infection) would be BCR mediated uptake of complexes that contain innate stimuli. However, the experiments performed by the authors in vitro instead seem to test the opposite scenario-where TLR activation precedes BCR activation and licenses B cells to respond to antigen. I think that a few additional in vitro experiments that address how the timing of exposure to these signals affects B1 responsiveness could at least begin to clarify some of these issues and thus improve the paper. For example, does prior engagement of the BCR in CD5+ B1 cells stabilize the inhibitory complex and prevent future licensing by TLRs? Also, can CD5+ B1 cells acquire antigen through BCR mediated endocytosis despite the blunting of signaling by CD5 (the answer to this may already be known)? If not, then how do they respond to TLR stimuli such as nucleic acids from dying cells that would typically be recognized in endosomes after BCR-mediated endocytosis? Or, is this licensing in vivo restricted mostly to TLR ligands that can either signal on the cell surface or be readily acquired by other mechanisms (pinocytosis)?

Reviewer #2:

In this manuscript, the authors showed that CD5 negative B1 cells were the source of IgM producing cells in the draining lymph nodes upon influenza infection. These CD5 negative B1 cells were originally CD5 positive B1 cells, and down-regulated CD5 expression during cell proliferation upon TLR stimulation. In vitro experiments showed that the down-regulation of CD5 was at the level of gene expression, and was not due to the expansion of contaminated originally CD5 negative B1 cells. The B1 cells secreted significantly higher amount of IgM after the down-regulation of CD5 when stimulated with TLR ligands. An in vivo approach using neonatal chimeric mice showed that CD5 positive B1 cells lose CD5 expression in the draining lymph nodes upon influenza or *Salmonella* infection, and these CD5 negative B1 cells secrete high amount of IgM. The molecular analysis of purified B1 cells showed that the down-regulation of CD5 enhanced the BCR signaling by releasing from the suppressive effect of CD5. Finally, TLR-/- CD5 positive B1 cells were not able to down-regulate CD5 and thus failed to elicit IgM response upon the influenza infection.

This study addressed a long-standing question how B1 cells secrete a large amount of IgM by rapidly responding to infections. Importantly, the authors demonstrated that IgM-producing CD5 negative B1b cells that were previously thought different origin from CD5 positive B1a cells, were phenotypically indistinguishable from TLR-stimulated B1a cells. This finding is exciting in the field of B1 cell biology, as it requests reconsideration for the interpretation of previous studies.

In general, the experiments were well done, but the data presentation and explanations could be improved. For example, some of the data could be moved to supplementary information to improve readability. In addition, some more elaboration would be required for a few of the data (see below). Although it is not strictly required, the hypothesis that TLR-signaling may "license" BCR-dependent reaction could be more deeply explored, as it is functionally important. For example, in Figure 7, what happens for the BCR-signaling in vitro if B1 cells are stimulated both with TLR-ligands and anti-IgM? Also, in Figure 8, does VDJ repertoire skew upon influenza infection due to the "licensed BCR reaction" in WT B1 cells, which may be disappeared in TLR-/- B1 cells? These data may add further value to this study.

Specific points:

Results section, paragraph four.

B-1(Igha) and B-2(Ighb) cells should read B-1(Ighb) and B-2(Igha) cells.

Results section, paragraph five.

“we expanded the analysis to include all IgM^b^-expressing (B-1 donor-derived) and IgM^a^- (recipient-derived) cells, regardless of expression of CD19 or other surface markers.” Please provide the gating strategy together with Figure 1F, as this gating is important to show the increased CD5-negative population.

Figure 2H-I: Please provide explanation for the purpose of this experiment. The authors concluded that "CD5+ B-1 cells lose CD5 surface and mRNA expression after in vitro LPS stimulation" based on this experiment. However, these experiments show that "CD5+B1 cells survive better, and also CD5+ and CD5- B1 cells proliferate similarly upon LPS stimulation". Thus, the data do not support the conclusion in the text. Please provide better explanation.

Results paragraph eight: Figure 3J-K should read Figure 2J-K.

Figure 3C-D: Why did the authors use Mtb lipids? Is this used as the alternative TLR4-ligand, which is in a different form from LPS? Please provide some explanation.

Figure 4F: Which chimeric mice were used for this analysis?

Figure 5: Please carefully check this figure and legend, because there are many typos here. This figure may be moved to supplementary information.

Title of the figure legend: “…and spleen after *S. typhimurium* infection” No data is shown for spleen.

Figure 5B: Graph label for the Y-axis: #CD5+ B-1 cells MedLN should read MesLN ?

Figure 5B: Why the numbers of CD5+ B-1 cell significantly reduced in 95% CD5+ chimera comparing with CD5+ chimera in mesenteric LN?

Figure 5B contain 2 graphs (MesLN and Peyer's patch), but the legend indicates different presentation: (B) MesLN and (C) Peyer's patch.

Third paragraph of subsection “CD5+ B-1 cells become CD5- IgM ASC in the Mesenteric LNs and Peyer’s Patches after *Salmonella typhimurium* infection”:

– Figure 5F is not provided.

– “Instead, we found OmpD-specific IgM secretion by B-2 derived plasmablasts.” should read “Instead, we found OmpD-specific IgM secretion by B-2 derived plasmablasts in mesenteric LN” (?).

– “Of note, the phenotype of B-2 derived plasmablasts is indistinguishable from that of "B-1b" cells (CD19lowCD45loIgM+CD43+ (Figure 5F)),…” Please provide the data to support this description (Figure 5F).

Reviewer #3:

In this interesting manuscript, Savage and colleagues provide evidence that innate-like CD5+ B1 cells respond to innate signals by production of IgM and differentiation into CD138+ plasmablasts in secondary lymphoid tissues. This was already known, however Savage and colleagues here demonstrate that CD5+ B1 cells are the actual TLR-responsive cells, and that TLR signaling activates these cells and results in downregulation of CD5. Potentially, this leads to disruption of CD5-IgM-BCR association and allows for subsequent BCR-medicated signaling, also the latter is not demonstrated.

This manuscript proposes a model in which CD5- B1 cells (B1b cells) are derived from CD5+ B1 cells (B1a cells), and reflect a post-activation state. This opposes the theory of 'division of labor' of B1a and B1b cells.

The manuscript is well written and the data are well presented. The overall message is interesting and adds to our knowledge of these innate-like B cells.

Questions/remarks:

1) As a general remark, the authors mostly focus on secondary lymphoid tissues, in which the differentiation into plasma cells probably occurs. Why do the authors not investigate the body cavities, where initial activation of B1 cells occurs? Downregulation of CD5 likely already occurs at this site.

2) What is the rationale of choosing the neonatal chimeric model as described in the Materials and methods for pleural cavity and mediastinal lymph node B1 cells, as the model is peritoneum-oriented (anti-IgM treatment and transfer of cells is i.p.)? This should be better explained.

3) The authors should include the transfer of CD5- cells as a control (now only 100% CD5+, 99% or 95% CD5+). Do CD5-B1 cells also upregulate CD138 upon TLR signaling?

4) Is there plasticity of CD5+ and CD5- B1 cells or is their fate set (in the pre-PC differentiated state)? mRNA (Figure 2) suggests transcriptional regulation of CD5 expression by TLR signaling (is this significant?). Is CD5 mRNA and surface expression restored upon removal of TLR agonists in vitro? This could be easily tested, and would add to the message of this manuscript.

5) Figure 6: the PLA data are difficult to interpret and may be redundant, as CD5 surface expression was already been shown to be lost upon TLR signals. This obviously affects the assay and it is thus questionable as of whether these results are useful. In addition, how can CD5 be detected in the PLA in CD5- B1 cells? (Figure 6A, panel 3).

6) Many plots are gated from IgM+cells (i.e. Figure 1A and Figure 8A): is it possible that the authors miss populations, as the B1 cells may have already undergone CSR, for example to IgA? Did they check for IgA+ or IgG+ B1-like cells? (dump-CD23-CD43+CD19^+^).

---

## [Author Response]

A key point in the discussion was the B cell-intrinsic nature of the TLR signaling in the other paper. We have requested a revision of both papers, and we would be grateful if you could liaise especially on this point to call out the complementarity.

We have inserted a more extensive discussion on the paper by Barton and colleagues in paragraphs two, three and five of the Discussion section.

Reviewer #1:In this manuscript, Baumgarth et al. describe their discovery and characterization of a novel mechanism by which innate signals can license B1 cells to respond to antigen receptor signaling and produce IgM. Specifically, they show that B cell-intrinsic TLR signaling can trigger the downregulation of the inhibitor of BCR signaling CD5, which effectively blocks antigen recognition-based activation of CD5+ B-1 cells. In so doing, the authors resolve multiple long-standing mysteries surrounding the mechanisms by which CD5+ B1 cells respond to antigen-dependent stimuli, overturn previously held views regarding the cellular origins of specific IgM responses, and uncover a defined mechanism by which B cells integrate innate and adaptive stimuli. The experiments that support these main conclusions are both elegant and, in my opinion, performed to an exceptionally high standard-e.g., the extensive use of neonatal B-1 allotype-chimeras enabled detailed and definitive examinations of the roles of B cell subsets in vivo. Overall, I have little to criticize regarding the veracity of the work and believe that this will be an important paper. My one remaining confusion has to do with the order of engagement of the innate versus adaptive stimuli. The authors speculate a bit about this in the discussion, but as far as I can tell don't directly address this experimentally. To me, the logical order of events under in vivo conditions (in response to self-derived debris or infection) would be BCR mediated uptake of complexes that contain innate stimuli. However, the experiments performed by the authors in vitro instead seem to test the opposite scenario-where TLR activation precedes BCR activation and licenses B cells to respond to antigen. I think that a few additional in vitro experiments that address how the timing of exposure to these signals affects B1 responsiveness could at least begin to clarify some of these issues and thus improve the paper. For example, does prior engagement of the BCR in CD5+ B1 cells stabilize the inhibitory complex and prevent future licensing by TLRs? Also, can CD5+ B1 cells acquire antigen through BCR mediated endocytosis despite the blunting of signaling by CD5 (the answer to this may already be known)? If not, then how do they respond to TLR stimuli such as nucleic acids from dying cells that would typically be recognized in endosomes after BCR-mediated endocytosis? Or, is this licensing in vivo restricted mostly to TLR ligands that can either signal on the cell surface or be readily acquired by other mechanisms (pinocytosis)?

We thank the reviewers for those comments. The question raised by this reviewer (and also reviewer #2) is very important and one we are currently considering how best to explore in the future. We agree that the order of stimulation (BCR/TLR) is not fully addressed in this manuscript. We expect additional extensive work is required to fully address this critical point which we believe to be outside the scope of this manuscript. However, in response to the comment, we have conducted additional experiments in vitro, now shown in a new Figure 7 and subsection “Initial stimulation through IgM-BCR suppresses subsequent activation of B-1 cells via TLR- stimulation” of the revised manuscript, in which we studied the effects of pulsed stimulation with one stimuli (either BCR or TLR for 2h and 24h followed by wash-out), followed by exposure to the second stimulus (for 70h and 48h, respectively). At 72h after initial stimulation we measured B-1 cell proliferation.

Stimulation for 2h with either ligand, followed by 70h stimulation with the other ligand showed no significant difference compared to 72h stimulation with the second stimulus alone (data not shown). However, when we exposed B-1 cells to the first stimulus for 24h and then the other stimulus for an additional 48h we found significant effects. Stimulation first with anti-IgM (BCR-signaling) and then CpG resulted in significant reductions in B-1 cell proliferation compared to stimulation only with CpG. In contrast, 24h stimulation with CpG followed by 48h stimulation via the BCR resulted in significant enhanced B-1 cell proliferation compared to BCR stimulation alone, however, still less than CpG only (Figure 7).

The data indicate that TLR stimulation must proceed BCR-stimulation for B-1 cells to be able to maximally respond to the innate stimulus. These data are consistent with the PLA experiments, which suggested strong BCR signaling-enhancing effects of TLR stimulation, while initial BCR stimulation enhanced the dissociation of the BCR with positive signaling components of the BCR complex.

How these findings relate to the findings by Barton and colleagues in the companion paper, in which they find effects of TLR-signaling on antigen specificity of the B cell response is now outlined in more detail in paragraphs two, three and five of the Discussion section.

Our current view is that the TLR-signal enhances potential responsiveness of B-1 cells in vivo, followed by a BCR-mediated signal that would then drive antigen-specific expansion of those cells that have already received a TLR signal. We are not aware of studies that have studied antigen uptake via BCR-internalization by B-1 cells. There are reports, however, that B-1 cells can uptake antigen via phagocytosis (Popi, Longo-Maugeri and Mariano, 2016), which would explain engagement of specific endosomal TLRs without first engagement and internalization of the BCR. This clearly requires further exploration, which we believe is outside the scope of this manuscript.

Importantly, for conventional B cells TLR-mediated stimulation may function quite differently, given their ability to vigorously respond to BCR-signaling alone. Here antigen-binding via the BCR would cause antigen/BCR-internalization and then exposure of the antigen to endosomal TLRs, enhancing antigen-specific responses.

Reviewer #2:In this manuscript, the authors showed that CD5 negative B1 cells were the source of IgM producing cells in the draining lymph nodes upon influenza infection. These CD5 negative B1 cells were originally CD5 positive B1 cells, and down-regulated CD5 expression during cell proliferation upon TLR stimulation. In vitro experiments showed that the down-regulation of CD5 was at the level of gene expression, and was not due to the expansion of contaminated originally CD5 negative B1 cells. The B1 cells secreted significantly higher amount of IgM after the down-regulation of CD5 when stimulated with TLR ligands. An in vivo approach using neonatal chimeric mice showed that CD5 positive B1 cells lose CD5 expression in the draining lymph nodes upon influenza or Salmonella infection, and these CD5 negative B1 cells secrete high amount of IgM. The molecular analysis of purified B1 cells showed that the down-regulation of CD5 enhanced the BCR signaling by releasing from the suppressive effect of CD5. Finally, TLR-/- CD5 positive B1 cells were not able to down-regulate CD5 and thus failed to elicit IgM response upon the influenza infection.This study addressed a long-standing question how B1 cells secrete a large amount of IgM by rapidly responding to infections. Importantly, the authors demonstrated that IgM-producing CD5 negative B1b cells that were previously thought different origin from CD5 positive B1a cells, were phenotypically indistinguishable from TLR-stimulated B1a cells. This finding is exciting in the field of B1 cell biology, as it requests reconsideration for the interpretation of previous studies.In general, the experiments were well done, but the data presentation and explanations could be improved. For example, some of the data could be moved to supplementary information to improve readability.

We thank the reviewer for these comments. We have now moved part of our data into two supplemental figures. Figure 2—figure supplement 1 contains data demonstrating that expansion of CD5- B-1 cells is not due to enhanced proliferation or survival by small frequencies of contaminating CD5- cultures at onset. Figure 5—figure supplement 1 now contains the data from the *Salmonella* infection experiments, which demonstrate that similar to what was observed after influenza infection, it is the CD5+ B-1 cell that responds to infection with downregulation of CD5 and differentiation to an IgM secreting cell. We hope that this together with changes listed below allows for an easier reading of the manuscript.

In addition, some more elaboration would be required for a few of the data (see below). Although it is not strictly required, the hypothesis that TLR-signaling may "license" BCR-dependent reaction could be more deeply explored, as it is functionally important. For example, in Figure 7, what happens for the BCR-signaling in vitro if B1 cells are stimulated both with TLR-ligands and anti-IgM?

As outlined in response to reviewer #1, who had similar questions, we now provide a new Figure 7 with in vitro data analyzing how the order of TLR and BCR signaling affects B-1 cell responses. We refer to the above response for details.

Also, in Figure 8, does VDJ repertoire skew upon influenza infection due to the "licensed BCR reaction" in WT B1 cells, which may be disappeared in TLR-/- B1 cells? These data may add further value to this study.

We thank the reviewer for that question. We agree that assessing the repertoire of B-1 cells in the mediastinal lymph nodes of influenza infected mice would be highly informative. Unfortunately, this is also a very challenging study to do, given the small numbers of B-1 cells that can be obtained from one mouse (about 2,000 – 5,000 total B-1 cells per animal). We hope that recent advances in single-cell RNA sequencing will help us to do this in the future. In terms of overall influenza-binding IgM-secreting cells, we previously published (Choi and Baumgarth, 2008) that the frequency of influenza-binding cells was similar in the MedLN between days 7 and 10 after infection and also similar compared to splenic B-1 cells prior to infection by ELISPOT analysis. Thus, we were unable to demonstrate influenza-specific clonal expansion. Thus, either the ELISPOT analysis lacked the necessary sensitivity to demonstrate expansion of influenza-binding B-1 cells, or the specificity of B-1 cells might be direct against altered self-antigens, rather than influenza virus and thus possibly triggered by DAMPS rather than PAMPS.

Specific points:Results section, paragraph four.B-1(Igha) and B-2(Ighb) cells should read B-1(Ighb) and B-2(Igha) cells.

We apologize for this mistake – it has been corrected in the revised manuscript.

Results section, paragraph five.“we expanded the analysis to include all IgM^b^-expressing (B-1 donor-derived) and IgM^a^- (recipient-derived) cells, regardless of expression of CD19 or other surface markers.” Please provide the gating strategy together with Figure 1F, as this gating is important to show the increased CD5-negative population.

In response we expanded the data shown to include all IgM^b^-expressing (B-1 donor-derived) and IgM^a^- (recipient-derived) cells, regardless of expression of CD19 or other surface markers. The data are added as a new part to Figure 1H.

Figure 2H-I: Please provide explanation for the purpose of this experiment. The authors concluded that "CD5+ B-1 cells lose CD5 surface and mRNA expression after in vitro LPS stimulation" based on this experiment. However, these experiments show that "CD5+B1 cells survive better, and also CD5+ and CD5- B1 cells proliferate similarly upon LPS stimulation". Thus, the data do not support the conclusion in the text. Please provide better explanation.

We are sorry for the confusion. We have moved subfigures 2F-I into the new Figure 2—figure supplement 1 to deconvolute the data. We have also reworded the text to make the purpose for the experiments clearer (subsection “CD5+ B-1 cells decrease CD5 expression after LPS stimulation in vitro”). In short, the question we attempted to address was whether the expanding population of CD5- cells we see emerge after stimulation of B-1 cells with TLRs could have arisen from small contaminations with CD5- cells that would survive better and/or proliferate stronger in response to the stimulus. We did not find that this was the case. Instead, if anything CD5- cells spiked into the cultures died faster and proliferated less than the CD5+ cells. From that we concluded that the increase in CD5^lo/-^ cells in the culture was due to the loss of CD5 expression of originally CD5+ cells.

Results paragraph eight: Figure 3J-K should read Figure 2J-K.

We have moved this data now into the new Figure 2—figure supplement 1 and reordered the figure.

Figure 3C-D: Why did the authors use Mtb lipids? Is this used as the alternative TLR4-ligand, which is in a different form from LPS? Please provide some explanation.

We now provide information and a reference (Basu, Shin and Jo, 2012) to indicate that the Mtb lipid provides a TLR 2 agonist.

Figure 4F: Which chimeric mice were used for this analysis?

We apologize for this not being clear. The entire figure was done with studies using the same sets of neonatal chimeras as outlined in the first sentence of the figure legend and then infected for 7 days with influenza A/PR8. To ensure this is clear we have now indicated this also in the Figure legend to Figure 4F.

Figure 5: Please carefully check this figure and legend, because there are many typos here. This figure may be moved to supplementary information.

We apologize for these errors. The figure has been carefully edited and moved to the supplemental information (Figure 2).

Title of the figure legend: “…and spleen after S. typhimurium infection” No data is shown for spleen.

We removed the reference to the spleen data.

Figure 5B: Graph label for the Y-axis: #CD5+ B-1 cells MedLN should read MesLN ?

We corrected the name of the lymph node.

Figure 5B: Why the numbers of CD5+ B-1 cell significantly reduced in 95% CD5+ chimera comparing with CD5+ chimera in mesenteric LN?

The chimeras always show a larger degree of variation in the sizes of their lymph nodes compared to that of non-chimeric mice. This is likely due to the fact that are set-up as a mix of males and females (it is difficult to determine the sex of a 1-day old mouse). B-1 cells are often somewhat more prevalent in females than in males. However, overall we found that the lymph nodes had similar levels of IgM secretion (5C). We therefore believe the variation to be not to be of biological significance (albeit reporting statistical significance) for the work done here. We now address this point in paragraph two of subsection “CD5+ B-1 cells become CD5- IgM ASC in the Mesenteric LNs and Peyer’s Patches after *Salmonella typhimurium* infection”.

Figure 5B contain 2 graphs (MesLN and Peyer's patch), but the legend indicates different presentation: (B) MesLN and (C) Peyer's patch.

We apologize for this mistake. The legend has been corrected.

Third paragraph of subsection “CD5+ B-1 cells become CD5- IgM ASC in the Mesenteric LNs and Peyer’s Patches after Salmonella typhimurium infection”:– Figure 5F is not provided.

The numbering of the figures has been updated.

– “Instead, we found OmpD-specific IgM secretion by B-2 derived plasmablasts.” should read “Instead, we found OmpD-specific IgM secretion by B-2 derived plasmablasts in mesenteric LN” (?).

We have altered the text as suggested.

– “Of note, the phenotype of B-2 derived plasmablasts is indistinguishable from that of "B-1b" cells (CD19lowCD45loIgM+CD43+ (Figure 5F)),…” Please provide the data to support this description (Figure 5F).

We have added a subfigure (Figure 5—figure supplement 1F) showing both IgM^b^+ and IgM^a^+ cells among the CD19lowCD43+CD138+ (IgM+) compartment in the MesLN.

Reviewer #3:In this interesting manuscript, Savage and colleagues provide evidence that innate-like CD5+ B1 cells respond to innate signals by production of IgM and differentiation into CD138+ plasmablasts in secondary lymphoid tissues. This was already known, however Savage and colleagues here demonstrate that CD5+ B1 cells are the actual TLR-responsive cells, and that TLR signaling activates these cells and results in downregulation of CD5. Potentially, this leads to disruption of CD5-IgM-BCR association and allows for subsequent BCR-medicated signaling, also the latter is not demonstrated.This manuscript proposes a model in which CD5- B1 cells (B1b cells) are derived from CD5+ B1 cells (B1a cells), and reflect a post-activation state. This opposes the theory of 'division of labor' of B1a and B1b cells.The manuscript is well written and the data are well presented. The overall message is interesting and adds to our knowledge of these innate-like B cells.Questions/remarks:1) As a general remark, the authors mostly focus on secondary lymphoid tissues, in which the differentiation into plasma cells probably occurs. Why do the authors not investigate the body cavities, where initial activation of B1 cells occurs?Downregulation of CD5 likely already occurs at this site.

We thank the reviewer for this comment. As we outlined in the Introduction of the manuscript, our previous work has shown that the cells entering the MedLN after influenza infection are nearly exclusively CD5+ B-1 cells (Choi and Baumgarth, 2008; Waffarn et al., 2015). The activation signal that traps these cells in the draining lymph node was shown by us to be the activation of the integrin CD11b, which was flipped into an active state via Type I IFNR signaling of pleural cavity B-1 cells by 2-days following influenza infection. At that timepoint the cells express CD5 at normal levels. While total numbers of CD5+ B-1 cells decline in the pleural cavity, the CD5- cells stay unaltered, suggesting that the CD5+ cells leave the pleural cavity before downregulating CD5 (Waffarn et al., 2015). We also generated CD5- only chimeras to demonstrate that CD5- B-1 cells cannot enter the lymph nodes (Choi et al., 2008). Based on these data, and the fact that IgM responses occur in the lymph nodes and not the body cavities, our focus has been on the secondary lymphoid tissues.

2) What is the rationale of choosing the neonatal chimeric model as described in the Materials and methods for pleural cavity and mediastinal lymph node B1 cells, as the model is peritoneum-oriented (anti-IgM treatment and transfer of cells is i.p.)? This should be better explained.

We apologize for this lack of clarity. The only way to inject a newborn mouse with B-1 cells is via the i.p. route. As was established in the original protocols we cited in the Materials and methods section and our own subsequent work (Lalor et al., 1989, Baumgarth et al., 1999, Baumgarth et al., 2000, Choi and Baumgarth, 2008, Choi et al., 2012, Waffarn et al., 2015 and Savage et al., 2017), subsequent to their injection i.p. B-1 cells in the neonate will expand into all compartments of the mouse, including both pleural and peritoneal cavities, spleen, bone marrow, all lymph nodes tested (mesenteric, mediastinal, inguinal, axillaris, cervical) as well as the gastrointestinal tract (Kroese et al., 1989) and lung (Baumgarth, unpublished). Thus, rather than seeing this chimera as a mouse with peritoneal cavity B-1 cells of a different allotype, the chimera has nearly all of its B-1 cell compartment exchanged for that of the donor. We have expanded the Materials and methods subsection “Chimera generation” to provide more information on these chimeras. We have also clarified the rationale for the use of the chimeras in the manuscript text in the first and fifth paragraphs of the revised Results section.

3) The authors should include the transfer of CD5- cells as a control (now only 100% CD5+, 99% or 95% CD5+). Do CD5-B1 cells also upregulate CD138 upon TLR signaling?

We thank the reviewer for this comment. We have previously conducted such a study and demonstrated that mice reconstituted with only CD5- B-1 cells have only very few MedLN B-1 cells and of those few that made it (about 200 B-1 cells in the entire lymph node), 65% of which expressed CD5, potential contaminants of our FACS-sorts, although we cannot exclude “reversion” to CD5+ status (Choi et al., 2008). For that reason we felt it was better in this manuscript to provide a range of mixes of CD5+ and CD5-, as this would expand on the data, by enabling us to correlate the secretion of IgM and CD5 expression with reconstitution frequencies. Importantly, our hypothesis to be tested was that CD5+ cells would lose CD5 expression. We now provide a more detailed description of that previous experiment and the rationale for the study, as conducted in subsection “CD5+ B-1 cells become CD5- IgM ASC in the MedLN after Influenza infection”.

4) Is there plasticity of CD5+ and CD5- B1 cells or is their fate set (in the pre-PC differentiated state)? mRNA (Figure 2) suggests transcriptional regulation of CD5 expression by TLR signaling (is this significant?). Is CD5 mRNA and surface expression restored upon removal of TLR agonists in vitro? This could be easily tested, and would add to the message of this manuscript.

We thank the reviewer for these comments. We have analyzed CD5 expression in cells that were pulsed with CpGs followed by 2-day exposure to anti-IgM stimulation or medium only. During that timeframe we did not see what appeared to be reexpression CD5 (not shown). The difficulty with interpretation of the data from those experiments is, however, that not all cells are stimulated to lose CD5 and it is not clear whether those expressing CD5 never lost its expression or potentially re-expressed it. Thus, although a “simple question” in principle, experimentally this is not quite as easy to address. Additional experiments that would test the effects of BCR-stimulation and/or removal of CpG stimulation followed by conditions that allowed B-1 cell survival without further stimulation would need to be developed, which we believe to be outside the scope of this manuscript.

5) Figure 6: the PLA data are difficult to interpret and may be redundant, as CD5 surface expression was already been shown to be lost upon TLR signals. This obviously affects the assay and it is thus questionable as of whether these results are useful. In addition, how can CD5 be detected in the PLA in CD5- B1 cells? (Figure 6A, panel 3).

We believe the PLA data to be of utmost important for interpretation of our studies. They demonstrate effects of CpG stimulation and BCR-engagement that are much more fine-grained than we can measure by surface staining only. What they demonstrate is that CD5 association with IgM is actually rapidly enhanced after BCR-signaling, while it is lost upon stimulation with CpGs – even at a time where the CD5 protein is still expressed on the cell surface. Thus, we see changes in the BCR complex configuration well before we see a loss of surface expression of CD5. Furthemore, we demonstrate effects of CpG stimulation on CD79a:Syk interaction, clearly linking TLR and BCR-complex signals.

To more clearly make this point we have rewritten the corresponding text in paragraph three of subsection “Changes in BCR signaling following innate activation of B-1 cells”. The very low frequencies expression of CD5 on the CD5- cell are presumably small impurities in the FACS-sort. We chose a figure that include one of those impurities for fullest disclosure, but as can be seen in the summary figure there frequencies are very low.

6) Many plots are gated from IgM+cells (i.e. Figure 1A and Figure 8A): is it possible that the authors miss populations, as the B1 cells may have already undergone CSR, for example to IgA? Did they check for IgA+ or IgG+ B1-like cells? (dump-CD23-CD43+CD19^+^)

We thank the reviewer for this comment. Previous work by us (Baumgarth et al., 1999) showed that for those antibodies for which we can discern allotypic differences (IgM, IgA, IgG1, IgG2a/c), the major contribution of B-1 cells was the generation of IgM. Hence we have focused on this response in our work and the goal of this manuscript is the study of IgM responses by B-1 cells. We agree with the reviewer, however, that B-1 cells may generate other isotypes. Of particular interest is IgG3. We showed data supporting strong contributions of IgG3 by B-1 cells in the steady state (Savage et al., 2017) and our colleagues in the Barton lab found previously that much microbial antibodies are of the IgG3 isotype that are generated by B-1 cells (Koch et al., 2016). The companion paper outlines their follow-up work (Kreuk et al., 2019). Unfortunately, IgG3 has no allotypic variation among mouse strains, and thus our chimera approach cannot distinguish B-1 from B-2-derived antibody secretion. In unpublished studies we looked at IgA production in the lung tissue before and after influenza infection and found the levels of IgA-production by B-1 cells to not be affected by the infection (Baumgarth, not published). However, a more detailed analysis on the MedLN might be warranted.